Corrected: Author correction

# High frequency temperature variability reduces the risk of coral bleaching

Aryan Safaie [1], Nyssa J. Silbiger [2,3], Timothy R. McClanahan[4], Geno Pawlak[5], Daniel J. Barshis[6], James L. Hench [7], Justin S. Rogers [8], Gareth J. Williams [9] & Kristen A. Davis [1]

Coral bleaching is the detrimental expulsion of algal symbionts from their cnidarian hosts, and predominantly occurs when corals are exposed to thermal stress. The incidence and severity of bleaching is often spatially heterogeneous within reef-scales (<1 km), and is therefore not predictable using conventional remote sensing products. Here, we systematically assess the relationship between in situ measurements of 20 environmental variables, along with seven remotely sensed SST thermal stress metrics, and 81 observed bleaching events at coral reef locations spanning five major reef regions globally. We find that high-frequency temperature variability (i.e., daily temperature range) was the most influential factor in predicting bleaching prevalence and had a mitigating effect, such that a 1 °C increase in daily temperature range would reduce the odds of more severe bleaching by a factor of 33. Our findings suggest that reefs with greater high-frequency temperature variability may represent particularly important opportunities to conserve coral ecosystems against the major threat posed by warming ocean temperatures.

[1] Department of Civil and Environmental Engineering, University of California, Irvine, CA 92697, USA. [2] Department of Biology, California State University, Northridge, 18111 Nordhoff Street, Northridge, CA 91330-8303, USA. [3] Department of Ecology and Evolutionary Biology, University of California, Irvine, CA 92697, USA. [4] Marine Programs, Wildlife Conservation Society, 2300 Southern Boulevard, Bronx, NY 10460, USA. [5] Department of Mechanical and Aerospace Engineering, University of California San Diego, 9500 Gilman Drive, MC0411, La Jolla, CA 92093, USA. [6] Department of Biological Sciences, Old Dominion University, Mills Godwin Building 110, Norfolk, VA 23529, USA. [7] Nicholas School of the Environment, Duke University, 135 Duke Marine Lab Road, Beaufort, NC 28516, USA. [8] Department of Civil and Environmental Engineering, Stanford University, 473 Via Ortega, Y2E2 Rm 126, Stanford, CA 94305, USA. [9] School of Ocean Sciences, Bangor University, Anglesey LL59 5AB, UK. Correspondence and requests for materials should be addressed to A.S. (email: safaiea@uci.edu) or to K.A.D. (email: davis@uci.edu)

Coral reef ecosystems provide subsistence nutrition, coastal protection, and revenue from tourism to hundreds of millions of people globally[1,2], and are valued at trillions of dollars annually[3]. Especially during recent years, coral reefs are increasingly threatened by accelerated rises in ocean temperatures owing to global warming[4–6]. Elevated seawater temperatures are the primary cause of mass coral bleaching[5,7], or the loss of pigmentation due to the collapse of the symbiotic relationship between the coral host and its endodermal dinoflagellate algae (zooxanthellae)[7,8]. Bleached corals are susceptible to disease[9] and reduced carbonate accretion[9,10], and prolonged bleaching will lead to mortality[5,11,12].

Thermal stress on corals and regional bleaching events are most often predicted by the magnitude and duration of remotely sensed sea surface temperatures (SSTs) above a fixed, locally defined average summer threshold temperature[5,8,13]. A conventionally used metric for quantifying these temperature anomalies is provided by the National Oceanic and Atmospheric Administration's (NOAA) Coral Reef Watch program, which has reported cumulative thermal stress on reefs twice a week since 1997[14]. Furthermore, bleaching predictions from remotely sensed temperatures can be improved by including SST-based calculations of interannual temperature variability[15,16] and coral sensitivity to thermal stress exposure[17]. However, the relatively coarse spatiotemporal resolution of the remotely sensed data prevents ensuing thermal stress quantifications from identifying the often observed significant spatial heterogeneity in bleaching that occurs within reef regions and individual reefs[18–20]. The response of reefs to temperature at these smaller spatial scales is complex and putatively depends on a combination of organism-level and reef-scale factors such as coral life-history strategies and stressor cotolerances[21], the history and duration of thermal stress exposure[22,23], the rate of change in seawater temperature[24,25], flow conditions[26], heterotrophic feeding[6], turbidity[27], and the intensity and history of exposure to solar radiation[28,29]. In turn, many of these environmental conditions are mediated by reef-scale factors such as waves[30], winds, tides[31], and daily heating and cooling[32].

Site-specific studies suggest that historical temperature variability within diurnal time scales affects corals' physiological tolerance[19,26,33,34] and performance[35] under thermal stress. For example, it has been theorized that corals located in areas characterized by large temperature fluctuations, such as reef flats or shallow lagoons, may be better acclimatized or adapted to thermal stress, and therefore more resistant to anomalous temperatures and bleaching, than corals in areas where temperatures are more stable, such as on reef crests or reef slopes[36–38]. Other studies have suggested that water temperatures in the weeks or months leading up to peak temperatures are critical in determining the coral physiological response. A recent analysis of experimentally heated corals from the Great Barrier Reef showed that bleaching and cell death responses were indeed lower when the thermal exposure included a moderate pre-stress followed by a short recovery period (i.e., a "protective temperature trajectory")[39]. Depending on intrinsic properties of coral physiology[40] such as energy reserves and algal phenotypic plasticity[41], pre-peak temperatures may either protect against or exacerbate bleaching at peak temperatures[41]. Taken together, a growing body of evidence thus suggests that historical temperature variability, and particularly, "high-frequency" temperature variability, which we define as occurring within diurnal or shorter periods, may play an important role in determining corals' physiological responses to thermal stress and thereby reef-scale vulnerability to bleaching. In turn, a better understanding of reef-scale bleaching risk factors could help coastal management efforts to identify natural refugia and may be important for the recovery of coral communities following a bleaching event[42].

Here, using a global suite of in situ data, we compare and assess the ability of 20 commonly used environmental variables and 7 remotely sensed variables to explain observed bleaching prevalence, testing the hypothesis that including high-frequency temperature variability as one of these model variables will lead to more accurate predictions. Analyzed data include records of in situ temperature time series at 118 reef locations from five major reef regions with sampling intervals of ≤3 h and continuous measurements of ≥1 year, as well as precise information on habitats and depths (Supplementary Data 1), along with 81 spatially and temporally coincident, quantitative coral bleaching observations (Supplementary Data 2). Each of the 81 bleaching observations was matched to its own spatiotemporally coincident temperature time series data, such that 46 of the 118 temperature time series were used in the subsequent bleaching analysis. Bleaching observations, which are most often reported as the average percent of colony or transect area bleached, were standardized to ordinal-valued "bleaching prevalence scores" (1: ≤10%; 2: 10−25%; 3: 25−50%; 4: >50% of reef area bleached), representing mild to pervasive bleaching, respectively (Methods). The influence of different factors on bleaching prevalence scores are evaluated by selecting covariates from a pool of 20 explanatory variables (depth, latitude, and 18 thermal metrics) grouped into 8 categories of metrics often used to predict bleaching (Table 1). In addition to these in situ variables, we also include 7 analogous and conventional remotely sensed SST thermal stress metrics (Table 1). After standardizing all covariates and fitting them to ordinal-valued bleaching prevalence scores using ordinal logistic regression (OLR) models (Methods), we conclude that high-frequency temperature variability, specifically the average daily temperature range (DTR) of the 30 days preceding a bleaching observation, is the most influential covariate in predicting the bleaching response, and serves to attenuate the prevalence of bleaching.

## Results

**Variation among in situ explanatory variables**. A principal components analysis (PCA) displays the projection for each site onto the 2D plane that accounts for the most variance in the 20 in situ explanatory variables (Fig. 1), and the locations of the loading vectors reveal how these explanatory variables relate to their respective groupings. The first principal component accounts for 44.2% of the variation in the explanatory variables, and is largely driven by high-frequency temperature variability and cumulative thermal stress.

**Spatiotemporal dependence of diurnal temperature variability**. The thermal metrics computed from temperature time series were highly variable across sites, but regardless of location and depth, all 118 time series show significant temperature variations in the high-frequency band (Supplementary Note 1; Supplementary Fig. 1), which we define as 0.727−4 cycles per day (cpd). Power spectra of temperature variations were calculated for each location, and the ratios of high-frequency band to seasonal band (0.012−0.143 cpd, or 1/7 to 1/84 days) variance in these spectra were used to characterize the relative importance of variance within the high-frequency band. This ratio correlates with the inverse of depth ($r = 0.381$, Student's $t$-test $p < 0.05$), indicating that the relative contribution of high-frequency variability to the variance within a temperature time series is stronger at shallower sites (Supplementary Fig. 2a). At back reef, reef flat, and reef slope habitats, these ratios were on average 1.83, 0.68, and 0.44, respectively, while across all locations, this ratio was 1.02 (Supplementary Fig. 2b). Furthermore, these ratios differed significantly among the three habitats (Kruskal-Wallis, $\chi^2 = 24.66$,

**Table 1 List of explanatory variables used in the ordinal logistic regression analysis**

| Category | Variable [Units] | Identifier | Description | Ref. |
|---|---|---|---|---|
| 1. Depth | Instrument depth [m] | depth | In situ water depth of instrument | |
| 2. Background Conditions | Latitude [DD] | lats | Latitude of instrument | |
| | Maximum Monthly Mean (MMM) [°C] | $MMM_{Total}$ | Maximum of monthly mean climatology from entire time series | 85 |
| | | MMM | Maximum of monthly mean climatology using data only before and during bleaching event | |
| | | $MMM_{4\,km}$ | Maximum of monthly mean climatology using 4 km weekly CoRTAD SST data | |
| | | $MMM_{Max}$ | Mean of maximum monthly SST from each year in climatological time period | 15 |
| 3. Cumulative Thermal Stress | Degree Heating Weeks (DHW) [°C-weeks] | $DHW_{90}$ | Trapezoidal integration of temperatures in excess of MMM+ 1 °C during 90 days preceding a bleaching event | 85 |
| | | $DHW_{30}$ | Trapezoidal integration of temperatures in excess of MMM+ 1 °C during 30 days preceding a bleaching event | |
| | | $DHW_{4\,km}$ | Degree heating week product from 4 km weekly CoRTAD SST data | |
| | Cumulative Summer Anomaly (CSA) [°C-days] | $CSA_{Total}$ | Trapezoidal integration of temperatures in excess of MMM+ 1 °C during all summer periods through entire time series | |
| | | $CSA_{Before}$ | Trapezoidal integration of temperatures in excess of MMM+ 1 °C during summer periods before and during a bleaching event | |
| | | $CSA_{During}$ | Trapezoidal integration of temperatures in excess of MMM+ 1 °C during summer of bleaching event | |
| 4. Acute Thermal Stress | Presence/absence of acute temperature anomaly [binary] | Acute1 | Binary value indicating whether any of the daily mean temperatures within 90 days preceeding a bleaching event exceeded MMM+ 1 °C | |
| | | $Acute1_{4\,km}$ | Acute1 computed using 4 km weekly CoRTAD SST data | |
| | | Acute2 | Binary value indicating whether any of the daily mean temperatures within 90 days preceeding a bleaching event exceeded MMM+ 2 °C | |
| | | $Acute2_{4\,km}$ | Acute2 computed using 4 km weekly CoRTAD SST data | |
| 5. Thermal Trajectory | Type of induced thermal tolerance prior to acute thermal stress, using twice-weekly averaged temperatures [ordinal] | TT | 0: No thermal stress (temperatures do not exceed MMM+ 2 °C within 90 days prior to survey date) 1: Protective Trajectory (temperatures exceed MMM, then have a recovery period below MMM for at least 10 days prior to exceeding MMM+ 2 °C) 2: Single Bleaching Trajectory (temperatures exceed both MMM and MMM+ 2 °C without a 10-day recovery period in between) 3: Repetitive Bleaching Trajectory (temperatures exceed MMM+ 2 °C in two peaks separated by 9 days) | 39 |
| 6. Heating Rate | Rate of spring-summer temperature change [°C/day] | $ROTC_{SS}$ | Mean rate of temperature change during spring and summer of all years | 25 |
| | | $ROTC_{90-4\,km}$ | Mean rate of temperature change during 90 days preceding a bleaching event using CoRTAD SST data | |
| | | $ROTC_{SS-4\,km}$ | Mean rate of temperature change during spring and summer of all years using CoRTAD SST data | |
| 7. High-Frequency Temperature Variability | Daily Temperature Range (DTR) [°C] | $DTR_{Total}$ | Mean DTR over entire time series | |
| | | $DTR_{SS}$ | Mean DTR of all spring and summer periods | |
| | | $DTR_{FW}$ | Mean DTR of all fall and winter periods | |
| | | $DTR_{90}$ | Mean DTR over 90 days preceding a bleaching event | |
| | | $DTR_{30}$ | Mean DTR over 30 days preceding a bleaching event | |
| 8. DTR Distribution Shape | Measure of shape of distribution of all DTR values w/in a time series [−] | kurtosis | Kurtosis of full time series of DTR values | |
| | | skewness | Skewness of full time series of DTR values | |

Variables are grouped according to eight categories representing different aspects of ecologically relevant environmental and temperature factors. Seasons were defined such that each season spanned three complete months, and austral and boreal summers were December through February and June through August, respectively

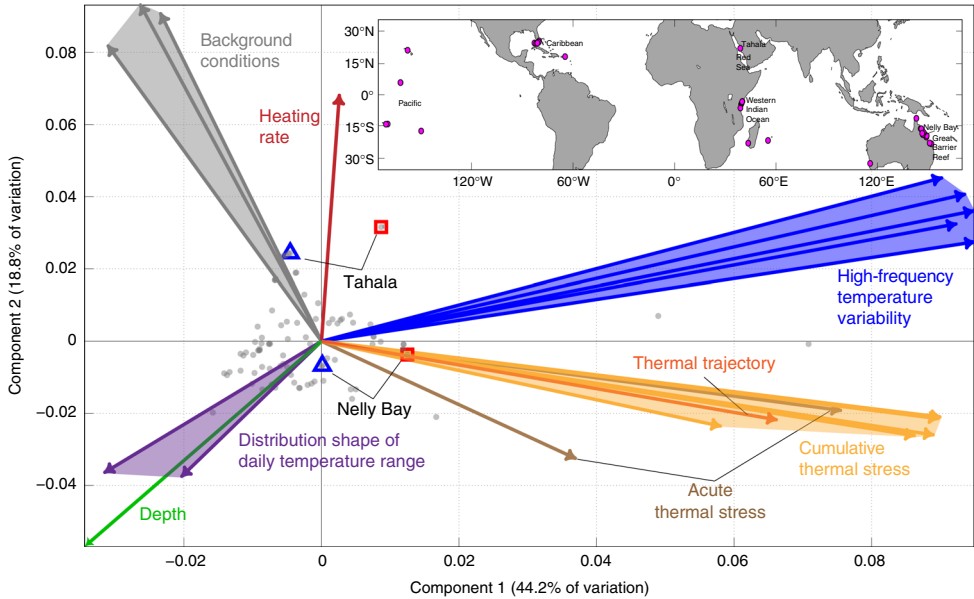

**Fig. 1** First two axes of variation of site-specific explanatory variables. Biplot of principal components analysis (PCA) showing the first two components (44.2% and 18.8%, respectively) that explain the majority of the variance in the matrix of 20 in situ explanatory variables (Table 1) used to explain bleaching prevalence. The light gray dots ("scores") each represent temperature time series associated with a distinct bleaching event at a given reef site. Gray dots that are close to each other have more similar temperature environments than dots further apart. The vectors are colored according to the categories described in Table 1. The time series inspected later in Fig. 4 are also indicated by red squares (Tahala and Nelly Bay shoreward habitats) and blue triangles (Tahala and Nelly Bay seaward habitats). The magenta circles in the inset map indicate the locations of all 118 in situ time series, with their associated reef regions labeled. The map was created using the MATLAB package "M_Map", created by Rich Pawlowicz under the license Copyright (c) 2014, Chad Greene. All rights reserved

df = 117, $p < 0.05$; Supplementary Fig. 2b). Although the magnitude of diurnal temperature fluctuations varies by location, the ubiquity and prominence of temperature variance in this frequency band indicated by these average ratios reflects the importance of some common physical forcing processes governing the flow and heating of reef waters, such as diurnal solar heating, tides, winds, and waves[31,32,43,44].

Power spectra of six representative time series from different reef regions (Fig. 2a) show a broad range of temperature variability from annual to hourly periods (see Supplementary Fig. 1 for other spectra). Yearly composites of mean water temperature and DTR (Fig. 2b) both show prominent seasonal cycles (Supplementary Note 2): the magnitude of daily temperature fluctuations was seasonally dependent (Kruskal-Wallis, $p < 0.01$) for 96% of reefs in our study (113 of 118 time series), with maximum DTRs occurring most often in spring and summer months (74% of time series, Supplementary Fig. 3), and minimum DTRs occurring most often in fall and winter months (also 74% of time series, Supplementary Fig. 3). On global scales (~$10^3$ km), latitudinal gradients in solar forcing drive variations in seasonal temperature patterns on reefs (Supplementary Fig. 4), but there is also considerable heterogeneity in thermal environments at reef-scales (~$10^2$ m) due to variation in depth and circulation[32,42,45]. The differences in thermal environments at reef-scales are often greatest in the high-frequency band (daily and tidal timescales; Fig. 2c). Dramatically different thermal environments can be found at locations separated by 10s or 100s of meters on a reef, as illustrated by 7-day temperature time series from various locations on the same island, or different habitats within a given reef (Fig. 2c). For example, during a week in November 2009, two sites in American Samoa that are separated by <2 km and at similar water depths experienced average DTRs of 1.78 and 0.51 °C (Fig. 2c, sites OF3 and OF5 respectively).

Differences in the distributions of DTRs that distinguish microclimates within a reef system (e.g., thermally variable

shoreward locations or thermally stable seaward ones) are reflected in the mean, skewness, and kurtosis of DTR values (Fig. 2d). Shallower and more shoreward sites have a peak in their DTR distributions corresponding to a larger DTR value, and furthermore, their distributions take on more extreme values than those from sites in deeper and more seaward locations. For example, at Heron Island in the Great Barrier Reef, the mean DTR of 4.23 °C on the reef flat was over three times as large as that of the reef slope (Fig. 2d). The implications of these different thermal microclimates for resistance to thermal stress and resilience to bleaching are discussed below.

**The effect of diurnal temperature variability on bleaching.**
Ordinal logistic regression ("logit") models were computed for all permutations of selecting at most one variable from each of the eight categories in Table 1 (a total of 10,367 models), with bleaching prevalence scores as the response variable. Corrected Akaike's Information Criterion ($AIC_C$) values were used to rank the logit models, where the model with the lowest $AIC_C$ value was ranked the highest (Fig. 3b). The model coefficients indicate the association of tested variables with bleaching prevalence score, such that positive coefficients indicate a "mitigating" effect on bleaching prevalence, and negative coefficients an "exacerbating" effect on bleaching prevalence.

"High-Frequency Temperature Variability" (Table 1) was used to capture temperature variability on diurnal and shorter time periods, a metric that is important for characterizing differential reef- and habitat-scale microclimates[19,32,46]. In the best model (Fig. 3a), high-frequency temperature variability, specifically the average DTR over the 30 days preceding a bleaching event ($DTR_{30}$, Table 1) was the most influential metric for predicting bleaching prevalence score, with greater daily temperature variability serving as a mitigating factor (Fig. 3b). Furthermore, among all models within 2 $AIC_C$ units of the highest ranked

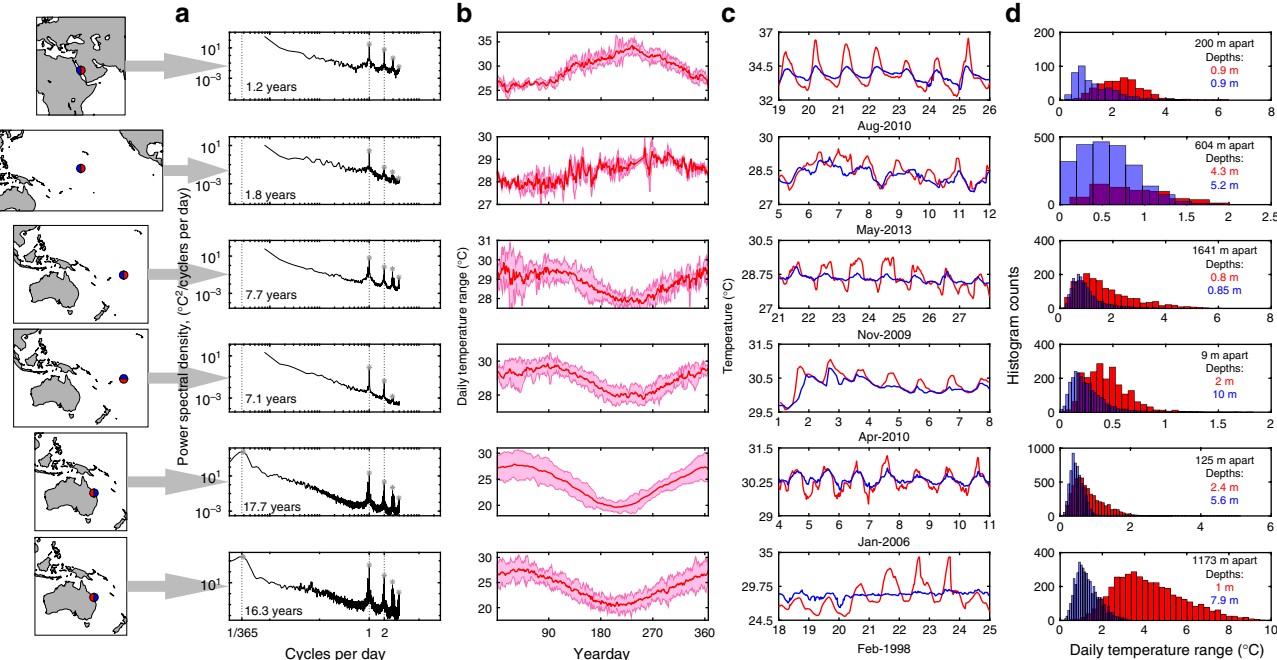

**Fig. 2** Temperature variability of six reef records. **a** Power spectra of temperature for TA3, P21, OF3, VT1, HW1, and HR1, with asterisks marking significant peaks, **b** yearly composites of mean daily temperatures and temperature ranges (red and pink shading respectively) for the same six time series in **a**, **c** 7-day trends in temperatures at two different habitats on the reef, and **d** histograms of daily temperature range at the same two habitats on each reef. In each case, reef locations are shown in maps on the left (for site information see Supplementary Data 1), the full duration of temperature records are indicated in **a**, and the great-circle distances between same-reef sites are indicated in **d**. The maps were created using the MATLAB package "M_Map", created by Rich Pawlowicz under the license Copyright (c) 2014, Chad Greene. All rights reserved

model (i.e. $\Delta AIC_C = AIC_C - \min(AIC_C) \leq 2$, Fig. 3a), high-frequency temperature variability was both the greatest mitigating factor of bleaching prevalence score and the most influential covariate—more influential than widely used metrics of acute and cumulative thermal stress by a factor of 2 and 3 times, respectively (Fig. 3c). Using globally averaged values of explanatory variables, our highest-ranked logit model (Fig. 3b) implies that, in native units, a 1 °C increase from the mean $DTR_{30}$ value would decrease the odds of more prevalent bleaching by a factor of 33. To standardize this, each unit increase in high-frequency temperature variability (i.e., $DTR_{30}$) would reduce the odds of more prevalent bleaching by a factor of $e^{2.66} = 14.3$. Contrasting this against a unit increase in cumulative thermal stress (i.e., $DHW_{30}$), which would only increase the odds of more prevalent bleaching by a factor of 2.6, highlights the dominant influence of diurnal temperature variability on reef-scale bleaching prevalence.

"Depth" (Table 1) was taken as the mean depth of the water temperature measurement, in meters below the surface, for each site, and is also representative of local water column depth as sensors were placed near the bed. Depth was the second-most effective predictor of bleaching prevalence (Fig. 3c), with deeper reefs less likely to experience pervasive bleaching. However, "depth" is also a proxy for other characteristics of the reef sites such as habitat (e.g., deeper forereefs and lagoons, shallow reef flats) and light intensity, which decays exponentially with depth. Although the logit models preclude significant collinearity of tested variables (Methods), corals at shallow depths may experience greater high-frequency temperature variability[45], although accounting for water flow can complicate this interpretation as it pertains to bleaching[26,47]. High-frequency temperature variability and depth may mitigate bleaching in complementary ways: habitats with greater high-frequency temperature variability, which are likely to be found at shallower

depths[45], may develop greater thermal tolerance[19,48], while deeper coral habitats, despite their propensity for milder diurnal temperature variability (outside of internal wave-influenced regions[49–51]), may serve as refuge areas resistant to the intrusion of hot water[25], perhaps facilitating recovery of coral cover following bleaching events[52].

"Background Conditions", "Cumulative Thermal Stress", and "Acute Thermal Stress" were the three explanatory variable categories largely suspected of exacerbating bleaching. "Background Conditions" (Table 1) consisted of the average summertime, or maximum monthly mean (MMM), temperature, but computed from our in situ time series data, as opposed to conventional remotely sensed SST data[14]. "Background Conditions" also included the latitude of the temperature logger, a variable that served as a proxy for unresolved oceanographic factors[53] related to the large-scale processes that influence climatologies. The "Cumulative Thermal Stress" category (Table 1) encompassed various methods for the computation of the magnitude and duration of acute in situ thermal stress exposure on reefs. Similar to the MMM, cumulative thermal stress is traditionally derived from remotely sensed SSTs and is among the most common metrics used to predict coral bleaching[25,33]. The "Acute Thermal Stress" category (Table 1) was included as a safeguard to differentiate sites with temperatures that may not have exceeded MMM+ 1 °C (i.e. no thermal stress) yet still experienced bleaching. Consistent with the well-established perspective that anomalously high temperatures are the primary cause of coral bleaching[7], among our highest ranked models, bleaching was most exacerbated by greater cumulative and acute thermal stress, and also, to a lesser degree, by increases in MMM temperature and heating rate[25]. "Heating Rate" (Table 1) was the average rate of change in spring to summer temperatures, which is believed to have a positive relationship with bleaching-induced tissue damage, and this time period has

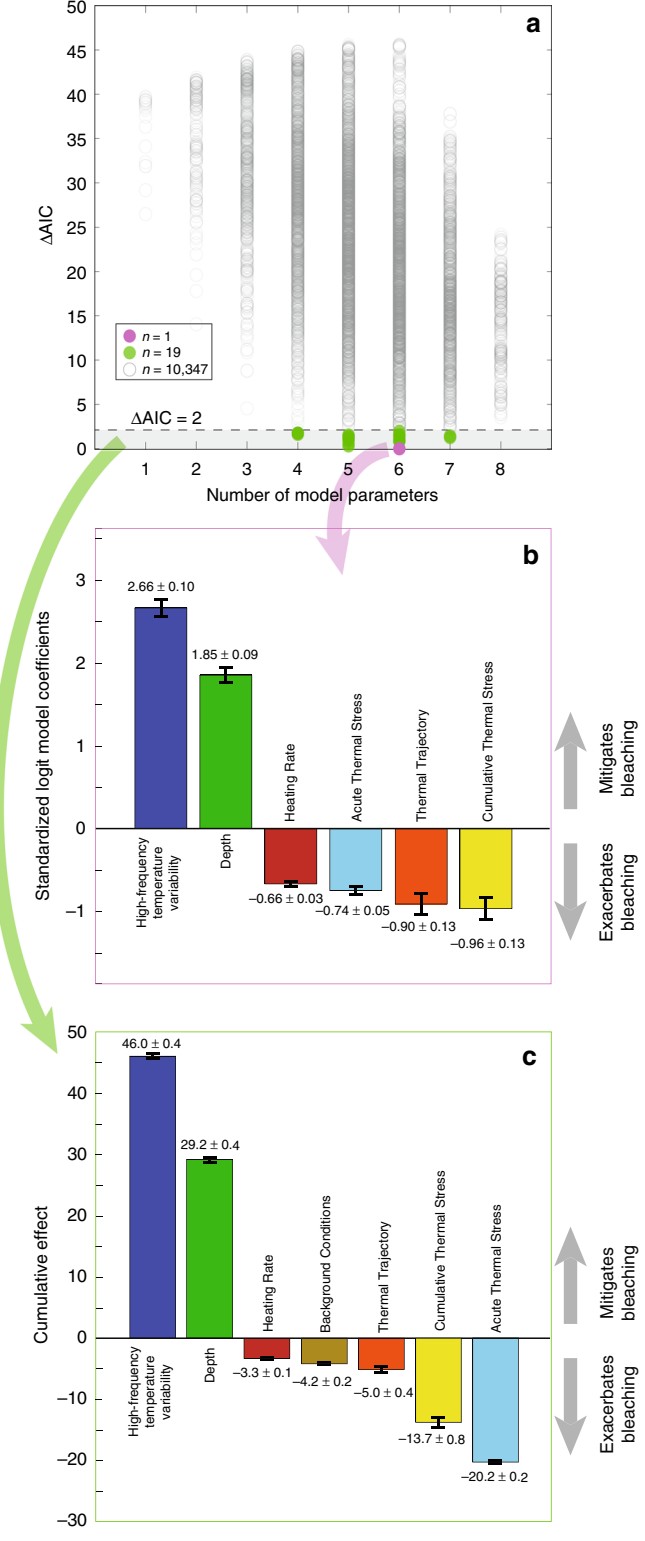

acute stress exposure. Although not as influential as the above variables, a no-stress or protective thermal trajectory (i.e., a pre-stress, sub-bleaching warming period, followed by a cooler recovery period) is more likely to result in lower bleaching prevalence than a single bleaching trajectory (temperatures that cross the bleaching threshold without a prior protective event) or a repetitive bleaching trajectory (Fig. 3b, c and Table 1). Finally, the "Shape of DTR Distribution" category (Table 1) was used to capture the skewness and kurtosis of DTR values derived from each time series to represent the symmetry and tail-density of DTR distributions. While these variables were not present in any of the highest ranked models, kurtosis and skewness of temperature time series have been associated with site-specific increased thermal tolerance[54].

To summarize the results of our highest-ranked logit model, we can examine how manipulating each covariate, while holding all others at their mean values, will change the probability of bleaching (Fig. 4). For example, a 0.88 °C decrease in high-frequency temperature variability ($DTR_{30}$) from its mean value would increase the probability of Category 4 bleaching from 12% to 75%, for a change of 63% (Fig. 4a), and a depth decrease of 5 m would increase this probability by 41% (Fig. 4b). Similarly, a 0.03 °C/day increase in $ROTC_{SS}$ from its mean value would increase the probability of Category 4 bleaching by 34% (Fig. 4c), and a 1 °C-weeks increase in $DHW_{30}$ would increase this probability by 44% (Fig. 4f).

To broaden the applicability of our conclusions, we repeated the OLR analysis, with the addition of remotely sensed SST-derived covariates, to determine how the results would differ from the in situ driven models. We obtained weekly 4 km resolution CoRTAD SST data[55], using the data pixels closest to the coordinates of our in situ loggers, and used this SST dataset to compute covariates within the Background Conditions, Acute and Cumulative Thermal Stress, and Heating Rate categories (Table 1). This resulted in an improved highest ranked model (Methods), that included six covariates, three of which (MMM, DHW, Rate of Temperature Change) were computed using the SST, as opposed to in situ, data (Fig. 5a). However, similar to the highest-ranked model fit to exclusively in situ data, covariates representing High-Frequency Temperature Variability, specifically $DTR_{30}$, and depth were again the dominant drivers of bleaching, and served as mitigating factors (Fig. 5a, b). Similarly, among covariates that exacerbated bleaching, Acute and Cumulative Thermal Stress provided the strongest influence (Fig. 5a, b), while Background Conditions ($MMM_{4 km}$, Table 1) represented a mild exacerbating effect. A notable difference occurring in these new models was the opposite effect Heating Rate had from before; whereas in the exclusively in situ models, Heating Rate exacerbated bleaching, these SST-based models imply stronger heating rates serve to mitigate bleaching. Ultimately, these

**Fig. 3** In situ explanatory variables of bleaching and their standardized logit coefficients with greatest predictive power. **a** $\Delta AIC_C$, computed as $AIC_C$ – min($AIC_C$), values of all 10,367 runs of an ordinal logistic regression model, where models within $\Delta AIC_C \leq 2$ (dashed line and gray shaded region) are statistically indistinguishable, of which there were 20. **b** The best model (i.e. $\Delta AIC_C = 0$) included six variables, of which high-frequency temperature variability was the absolute most influential and also greatest mitigating factor to bleaching prevalence. **c** Summing across 20 indistinguishably good models (i.e. within $\Delta AIC_C \leq 2$), high-frequency temperature variability was consistently most influential. Variable categories are shown in Table 1. Delete-1 jackknife standard error bars are shown in (**b**), while the standard error bars shown in (**c**) were obtained by summing in quadrature the individual standard errors from each of the 20 models computed after delete-1 jackknife resampling

been shown to be crucial for determining the fate of corals to summertime bleaching susceptibility[25]. The "Thermal Trajectory" (Table 1) category followed the methodology of a previous study that highlighted the role of protective warm, pre-stress temperatures as being important for resilience to bleaching from intense acute stress temperature events[39]. Our results reinforce recent findings that a reef's thermal trajectory is a significant predictor of bleaching prevalence[39] (Fig. 3c), with thermal tolerance conferred by exposure to a protective, sub-lethal bleaching stress prior to

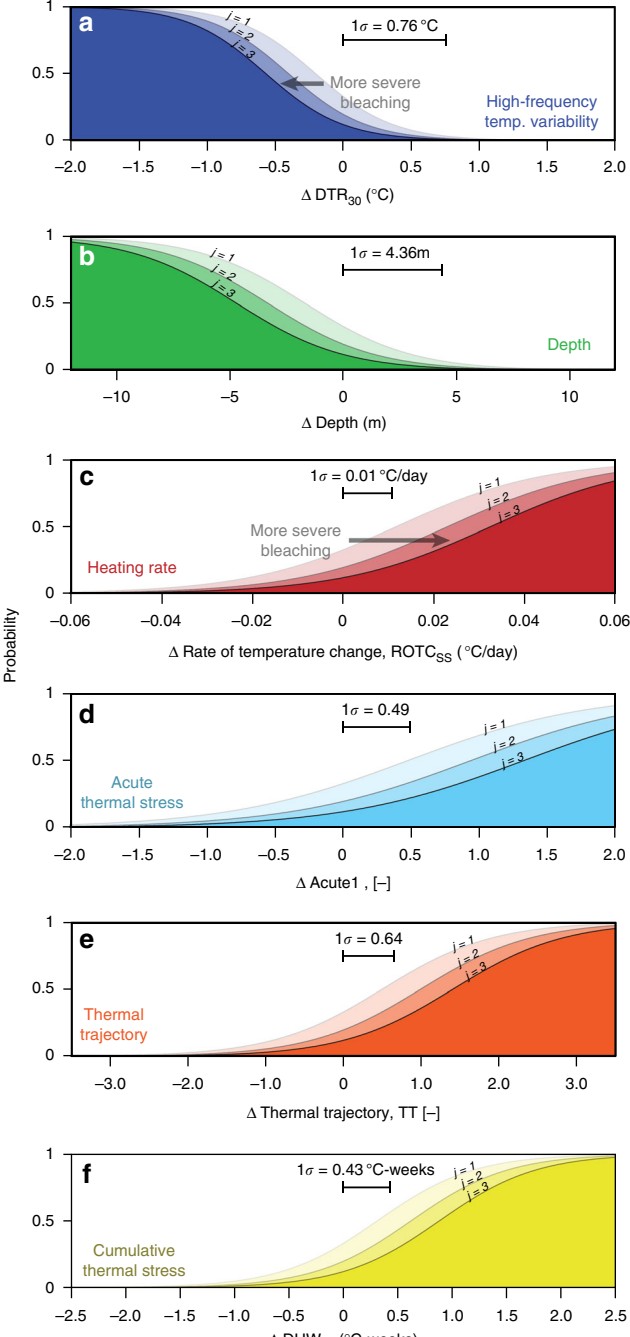

**Fig. 4** Influence of each in situ covariate on bleaching. Using the covariates from the highest-ranked logit model, the probability of observing bleaching prevalence greater than the *j*th category is plotted against changes in each covariate from their respective mean values (where 0 corresponds to the mean value), while keeping all other covariates at their mean values. Bleaching prevalence categories are defined as 1: ≤10%; 2: 10−25%; 3: 25−50%; 4: >50% of reef area bleached. Highest-ranked model covariates include: (**a**) High-frequency temperature variability ($DTR_{30}$), (**b**) Depth, (**c**) Heating Rate ($ROTC_{SS}$), (**d**) Acute Thermal Stress (Acute1), (**e**) Thermal Trajectory (TT), and (**f**) Cumulative Thermal Stress ($DHW_{30}$). Standard deviations for each covariate within our data set are also indicated

SST-based OLR models indicate that upon consideration of the consistent importance of DTR to bleaching, a globally available remotely sensed metric for diurnal temperature variability would be valuable for improved bleaching predictions.

**Specific reef cases**. Our results reveal the importance of high-frequency temperature variability at locations worldwide, but include reef-scale observations where such variability influences bleaching prevalence of corals in different locations of the same reef during the same bleaching event[32,46]. Here, we present two such case studies: one from Tahala Reef, a platform reef in the central Red Sea, and another from Nelly Bay, a fringing reef in the Great Barrier Reef in Australia (Fig. 6). These sites were chosen due to the availability of additional meteorological data[32,56] at these reefs. At each location, temperature time series (Fig. 6a, b) for both a seaward and a shoreward location show that, whereas low-frequency variations in water temperature are often very similar over reef-scales (Fig. 6c, d), high-frequency variations may be quite distinct (Fig. 6e, f). In these cases, bleaching events were more widespread and severe at the seaward locations where DTRs were smaller (Fig. 6a, b), consistent with our best logit models.

## Discussion

For corals, a shift in thermal tolerance can occur due to adaptation of the coral animal or algal symbionts through natural selection of heat-tolerant lineages[57,58], or physiological acclimation through the expression of heat shock proteins and regulation of apoptosis (i.e., programmed cell death)[23,36]. As discussed, recent work highlights the importance of short-term temperature history (daily-weekly periods) for coral acclimatization to higher temperatures[19], such that corals subject to warmer than average temperatures prior to thermal stress may exhibit a greater tolerance to acute temperature stress[23]. In the context of these studies and in keeping with other site-specific and experimental studies[19,26,45,47,59], our results suggest that temperature fluctuations on daily or tidal timescales are often sufficient to expose corals to temperatures high enough to encourage greater tolerance (via acclimation or adaptation) to thermal stress, but for time periods short enough to avoid mortality[19,48]. Further, our results establish that the resistance of corals located in areas of high-frequency temperature variability to bleaching occurs in reef regions throughout the world. While we lack sufficient species-level data, we fully acknowledge that intrinsic coral properties[40], differences in reef-scale community compositions[56], and taxonomic susceptibility[26] are likely to influence heterogeneous reef-scale bleaching responses, and an improvement to our model framework would include species-level covariates.

Our results also demonstrate the potential to both improve predictions of bleaching and anticipate heterogeneous patterns in bleaching prevalence at reef-scales, considering that beyond mean values, accounting for variability in temperature regimes yields better predictions of organismal responses to anomalous environmental events[60]. Although SSTs from satellite remote sensing are not yet available at the spatiotemporal resolution required to calculate reef-scale high-frequency temperature variability, observational work on a range of reef structures suggests that it may be possible to predict reef-scale thermal environments using relatively simple hydrodynamic models given readily available bathymetry and basic hydrographic data such as tidal range, wave height, and offshore mean SST[30,32,43]. While we did not assess other biogeochemical parameters, it is worth noting that the same physical circulation that drives spatially variable thermal environments, in locations with an active benthic community, can also create dynamic oxygen, pH, and nutrient environments[61,62].

Urgent global efforts at reducing anthropogenic greenhouse gas emissions must remain a priority for reef preservation, due to the acute thermal stress that is now arising from global warming projections on reefs[5,63]. However, combating the effects of local stressors on reefs through conservation tools such as marine protected areas is likely to increase the chances of reef persistence

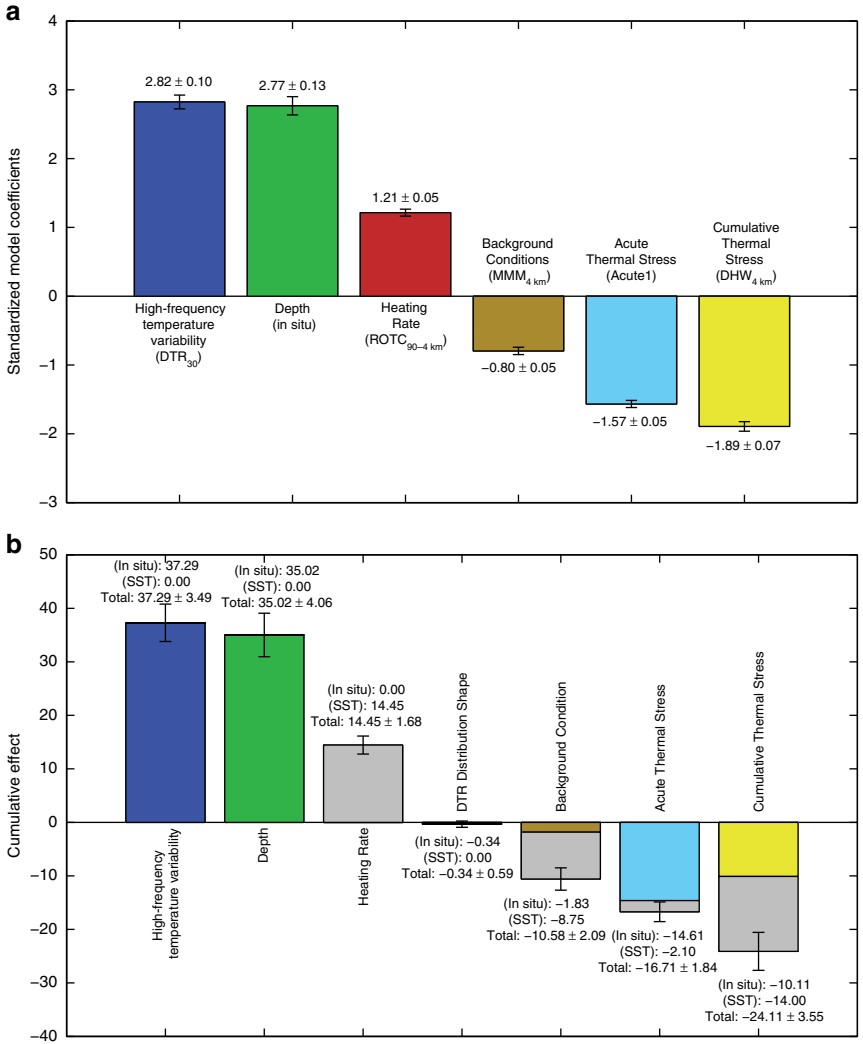

**Fig. 5** Remotely sensed SST OLR results. **a** Parameter estimates for standardized model coefficients of the covariates used in the highest-ranked OLR model when weekly 4 km CoRTAD SST-based variables are added to the pool of possible covariates; standard error bars were computed from delete-1 jackknife resampling. **b** The summation of the standardized covariate coefficients grouped by category from the highest-ranked models when including CoRTAD SST-based covariates; the standard error bars shown are obtained by summing in quadrature the individual standard errors computed after delete-1 jackknife resampling. The remotely sensed SST-based covariate contribution to each Cumulative Effect is colored gray

through future warming, as well as to facilitate recovery following a bleaching event[2,52,64,65]. Warming ocean temperatures are projected to result in annual severe bleaching regimes by the middle of this century, with spatial variability in the onset of these events on the order of ±10 years[4]. Considering our results in the context of this inevitable and persistent acute thermal stress, focusing management efforts on the more resistant reef locations that also experience delayed onsets in annual severe bleaching would maximize the likelihood that at least some healthy reefs will exist in the future.

## Methods

**Data synthesis**. Water temperature time series from 118 locations, representing five major ocean basins where tropical coral reefs are found (Western Indian Ocean, Pacific Ocean, Caribbean Sea, Great Barrier Reef, and Red Sea), were obtained from existing records of in situ temperature data. Many of these time series were obtained directly from the researchers (Supplementary Data 1), or from publicly available databases including the Australian Institute of Marine Science (AIMS), the National Data Buoy Center (NDBC), and the Florida Institute of Oceanography (FIO). However, time series within our dataset were also selected to match precise reef locations that had sufficiently documented bleaching events, while also containing as long and consistently sampled records as possible. Site names and three-letter codes, locations, depths, instrument descriptions, and

additional information for each time series are listed in Supplementary Data 1, which also lists the source for each time series. Water temperature records included in this analysis spanned at least 12 months in duration, with a sampling interval less than or equal to 3 h. In cases where instrument substitution resulted in varying sampling intervals, time series were sub-sampled to the largest of these intervals, or, in rare cases, interpolated to remain below a 3 h sampling interval. The temperature time series data used in this study originate from instruments that were calibrated using varying methodologies, such as by placing loggers together at one location and comparing recorded temperatures with a reference temperature dataset, ice bath calibration, or multiplying raw field recorded temperatures by normalized logger calibration coefficients. Our analysis is largely based on relative temperature variations, however, so that absolute temperature accuracy will not affect the results presented here. In an effort to examine how representative our sample temperature time series data was, we used one-sample *t* tests to compare the overall means and extremes of our data to a global temperature data set taken from nearly 1000 reef locations[66]. From these tests, we cannot conclude any significant differences ($\alpha = 0.05$) between our time series data and this larger global data set. All data analyses were done using MATLAB 7.14 (The Mathworks, Natick, MA, USA).

**Spectral analysis**. Power spectral density (PSD) estimates were computed for each temperature time series. First, if necessary, temperature time series were resampled or linearly interpolated to maintain a constant sampling interval, chosen to be 3 h so as to resolve spectral frequencies of up to 4 cpd. In order to examine temperature variability for a broad range of frequencies spanning annual to diurnal and shorter periods, a PSD was calculated as follows: time series greater than or

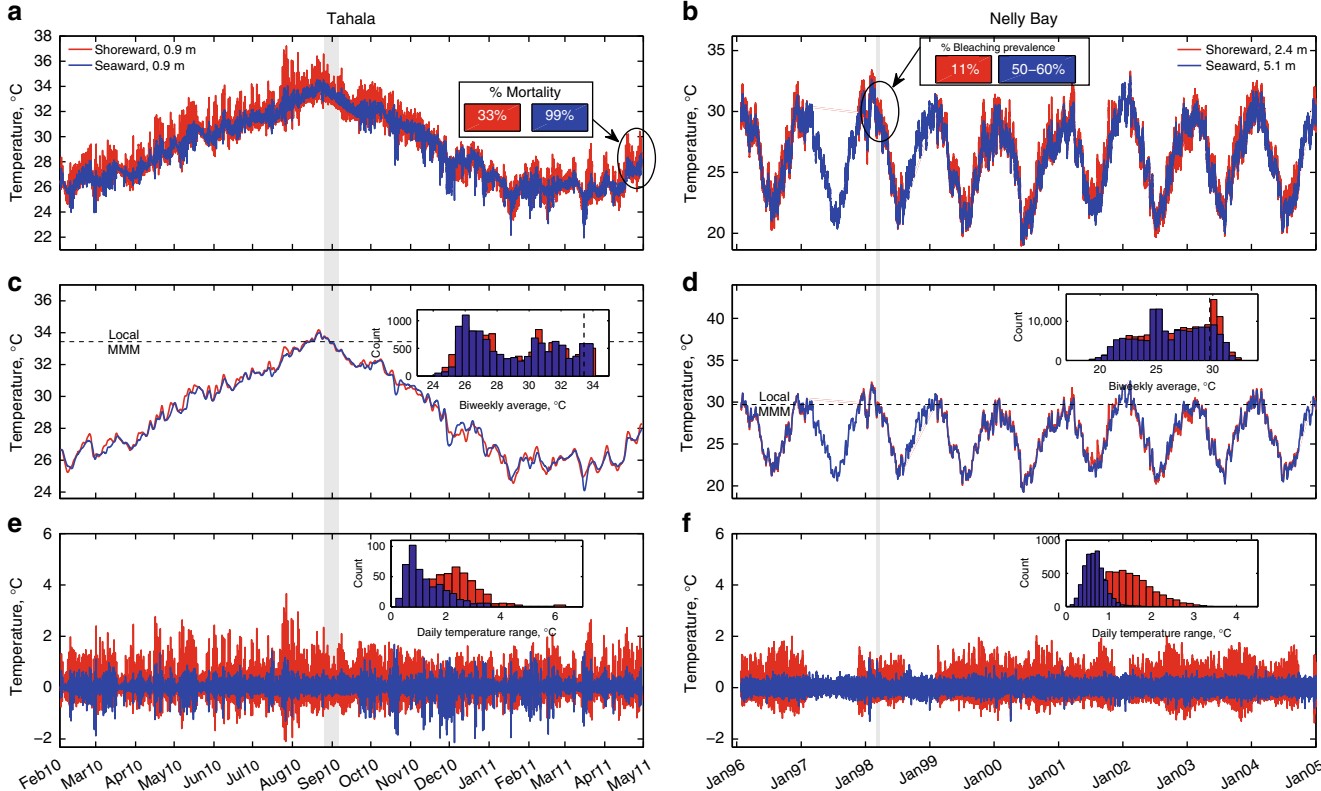

**Fig. 6** Same-reef case studies. **a** Temperature time series taken from the wave-exposed (blue) and wave-protected (red) edges of the Tahala reef platform in the Red Sea, which are separated by ~200 m. The percentage of observed mortality, which was associated with the bleaching event in September 2010[46,84], is indicated for each corresponding platform edge. **c** 2-week low-pass filtered time series of the raw Tahala temperature data, with the Maximum Monthly Mean (MMM) temperature calculated using the in situ data for each time series. **e** 33-h high-pass filtered time series of the raw Tahala temperature data, with a histogram of the Daily Temperature Range (DTR) values for each time series. **b**, **d**, **f** Analogous versions of **a**, **c**, and **e**, respectively, but for the Nelly Bay reef flat (red) and reef slope (blue) habitats in the Great Barrier Reef, separated by ~122 m. Bleaching prevalence as proportions of belt transects[56] are also indicated in **b**. The gray bars highlight the approximate periods of reported bleaching events

equal to 10 years in duration were divided into 3-year sections, while all others were divided into 4-month sections. Sections were overlapped by 50% and windowed with a Hamming function before spectra for each section were calculated, which were then ensemble averaged to obtain the PSD estimate. The statistical significance ($\alpha = 0.05$) of observed spectral peaks was ascertained by comparison with the upper confidence level of a background red noise fit to the spectrum[67]. We empirically defined annual, seasonal, and diurnal frequency bands as 0.00185 to 0.0111, 0.0119 to 0.143, and 0.727 to 1.333 cycles per day (cpd), respectively. Band variance was computed using trapezoidal integration of PSD values within each respective frequency range, and ratios of high-frequency to seasonal band variances among habitat types ("Back Reef", "Reef Flat", "Reef Slope") were compared using a Kruskal−Wallis test (Supplementary Fig. 2b).

**Spatiotemporal variability in water temperature**. To quantify the magnitude of diurnal patterns of heating and cooling, a DTR was calculated as the difference between maximum and minimum temperatures for each day of each time series. Temporal variations in DTR values were examined in multiple ways. First, DTRs were composite-averaged for yeardays 1−366, and based on evident seasonal DTR variability from panel **b** of Fig. 2, a non-parametric Kruskal−Wallis test was used to assess the seasonal dependence of DTR distributions[68]. Seasons were defined such that each season spanned 3 complete months, and austral and boreal summers were December through February and June through August, respectively. Temperature metrics are summarized in Table 1. Large-scale spatial patterns in water temperature variability relative to latitude were also characterized; annual temperature ranges, calculated as the range of monthly mean temperatures, were compared against latitude for all sites. Variance in high-frequency (33 to 6 h periods) and seasonal (7 to 84-day periods) spectral bands were computed via integration of their respective power spectral densities, and the ratios of high-frequency to seasonal variance for all time series were compared by the habitat from which each time series was recorded. Habitats were divided into three groups: (i) back reefs and back reef lagoons (labeled as "BR"), (ii) reef flats and non-back reef lagoons ("RF"), and (iii) reef crests, reef slopes, forereefs, and anything further offshore ("RS").

**Bleaching data synthesis**. Coral bleaching observations (81 events) that corresponded in time and location with temperature data (Supplementary Data 2) were obtained from a variety of sources, but primarily from peer-reviewed publications. Bleaching reports were based upon various quantification schemes, but some common methods included recording bleaching as: (i) a proportion of transect area[69], (ii) severity among different colonies of different coral species[70], and (iii) prevalence categories based on aerial (and ground-verified) surveys[22]. In publications that reported percentages of colony bleaching within different coloration and paling categories, we used the weighted average of the different percentages within each category, though the majority of bleaching records naturally translated into our four bleaching prevalence scores. To aggregate or standardize the bleaching reports for use in this study, we defined a bleaching response variable in terms of percentage of spatial area bleached, and we therefore assigned the following categories as ordinal values of bleaching prevalence score: 1: "bleaching prevalence" ≤ 10% ($n = 48$); 2: 10% < "bleaching prevalence" ≤ 25% ($n = 5$); 3: 25% < "bleaching prevalence" ≤ 50% ($n = 6$); 4: "bleaching prevalence" > 50% ($n = 22$). Note that bleaching events of 0 (reports of no bleaching observed or negligibly mild paling) are binned into bleaching prevalence score 1, to provide a conservative grouping for mild bleaching events, seasonal patterns of discoloration and variation in zooxanthellae densities[71], unresolved "background bleaching" levels[72], and observation errors. As opposed to continuous interval variables, ordinal variables represent categorical values that can be ranked and have a natural ordering to them, offering the advantage of creating bins for ranges of values.

**Explanatory variables**. To assess the influence of explanatory environmental variables, including high-frequency temperature variability, on bleaching response, we performed a multivariate statistical analysis of the observed bleaching events. As there are multiple aspects of thermal stress and environmental conditions that may explain the bleaching response, we selected 20 experimental variables for the ordinal regression (defined in Table 1), organized into eight broad categories: (1) Depth, (2) Background Oceanographic Conditions, (3) Cumulative Thermal Stress, (4) Acute Thermal Stress, (5) Thermal Trajectory, (6) Heating Rate, (7) High-Frequency Temperature Variability, and (8) Shape of the Distribution of HF

**Temperature Variability.** We lacked sufficient habitat (i.e., reef flat, reef crest, reef slope, etc.) information at which many loggers were placed, and therefore did not include habitat as an explanatory variable. Environmental variables in the Cumulative Thermal Stress category were calculated from in situ temperature measurements rather than the remotely sensed National Oceanic and Atmospheric Administration's Coral Reef Watch (NOAA CRW) products because many of our logger locations were outside of the coverage areas of the CRW Virtual Stations, and furthermore, as the first suite of CRW products was released in 2000, they do not include temperature data corresponding to the 1998 global bleaching event. The explanatory variables we use here are commonly appended with a subscript to indicate the period of time, relative to the bleaching observation date, used to calculate the metric. For example, the subscript "SS" denotes the spring to summer period, and the subscript "30" denotes the 30 days preceding (and including) a bleaching observation.

**Principal components analysis.** Principal components analysis was performed to examine the spatial structure of the environmental forcings and determine the association between independent variables within each of the eight categories (Fig. 1). The first two PC axes accounted for 44.2 and 18.8% of the variance within the matrix of independent variables. The magnitude and orientation of the loading vectors indicate the importance of each parameter in describing the variance of the PCA components.

**Computation of thermal trajectory and acute stress variables.** An ordinal-valued Thermal Trajectory[39] variable was included as an independent variable to assess the degree to which environmental conditions confer thermal tolerance. The calculation of this was as follows: first the MMM and MMM + 2 °C (the latter quantity is referred to as the "local bleaching threshold") were computed for a given time series. Then the 33-h low pass filtered time series for the 90 days preceding a bleaching event was inspected to determine the type of Thermal Trajectory. If temperatures exceeded the MMM, then fell below the MMM for a required 10-day "recovery period" before proceeding to exceed the local bleaching threshold, a Protective Trajectory with an ordinal value of 1 was recorded. If temperatures increased from below the MMM to above the local bleaching threshold, without a 10-day recovery period, a Single Bleaching Trajectory with a value of 2 was recorded. If temperatures exceeded the local bleaching threshold at least twice, with a required 9-day recovery period between threshold exceedances, a Repetitive Bleaching Trajectory with an ordinal value of 3 was assigned. Finally, if temperatures did not exceed the local bleaching threshold, an ordinal value of 0 corresponding to no thermal stress was assigned. Justification of our ordinal-value scheme comes from an analysis of experimentally heated corals from the Great Barrier Reef, whereby in the face of thermal stress, corals with a Protective Trajectory experienced localized cell death of approximately 30%, while that of corals under Single and Repetitive Bleaching Trajectories was approximately 60% and 70%, respectively[39]. Furthermore, experimental results showed that corals under a Protective Trajectory maintained significantly greater symbiont density than those under Single and Repetitive Bleaching Trajectories, and hence the ordinal scores from 0 to 3 aptly account for the monotonic nature of coral tissue detriment associated with no heat stress, a Protective Trajectory, a Single Bleaching Trajectory, or a Repetitive Bleaching Trajectory, respectively. The "No Thermal Stress" trajectory corresponded to ~0% cell death and greater symbiont density than a Protective Trajectory.

The Acute Thermal Stress category was composed of two binary variables to indicate the presence/absence of acute thermal stress. The calculation of this was as follows: first the MMM + 1 °C and MMM + 2 °C were computed for a given time series, then the daily mean temperatures within 90 days before a bleaching event were inspected to determine if temperatures exceeded MMM + 1 °C (in which case Acute1 would equal 1) and MMM + 2 °C (in which case Acute2 would equal 1).

**Ordinal logistic regression.** Using the eight explanatory variable categories described above, and bleaching prevalence scores from 1 to 4 as the response variable, we performed an ordinal logistic regression analysis to determine how the relative log odds of a given bleaching prevalence score depends on the interactions among the explanatory variables. Ordinal regression models have been adeptly used in ecological studies where data are often present as semi-quantitative variables in which relative differences between values are of importance[73,74]. Furthermore, logit functions have been previously implemented to predict the presence/absence of bleaching using gridded remote-sensed data[72,75], or to explain the influence of a range of environmental and coral physiological factors on reef ecosystem response following a disturbance[52]. These logit models are multivariate extensions of generalized linear regression models[76], providing parameter estimates via maximum likelihood estimation (MLE) to model the relative log odds of, for our purposes, observing one bleaching prevalence score or less versus observing the remaining greater bleaching prevalence scores:

$$\ln\left(\frac{P(y_i \le j)}{P(y_i > j)}\right) = C_j + B_1 z_{i1} + \cdots + B_p z_{ip}. \quad (1)$$

Here $i$ indexes each of $N$ observations, with bleaching observation $y_i$, and the left-hand side quantity is referred to as the logit of the probability of observing bleaching prevalence score $j$ or lower, for $j = 1, 2,$ or 3 (observations with bleaching prevalence scores of 4 contribute to the regression through calculation of the log-odds). Note that the odds are defined as the ratio of the probability of an event occurring to the probability of the event not occurring, which is exactly the ratio inside the natural logarithm. Each $C_j$ is an MLE-computed model intercept, and each $B_k$ is the MLE coefficient corresponding to each standardized independent variable $z_{ik}$, for $k = 1, \ldots, p$, where $p$ is the number of independent variables used in a given model. A fundamental component of this model is the assumption of proportional odds, or parallel regression, which implies $B_k$ values are independent of the logit level $j$. The validity of this parallel regression assumption was ascertained using Brant's Wald test[77], as well as a likelihood ratio test ($\alpha = 0.05$).

As each time series was the sole source of its explanatory variables, we can expect many of these variables to be correlated with each other (multicollinear). If left unaccounted for, multicollinearity obscures the interpretation of the explanatory variables and their coefficients, and may decrease the statistical power of the logit analysis[78]. The degree of multicollinearity among the 20 explanatory variables used in the logistic regression was assessed by calculating the condition indices and variance-decomposition proportions[79] of the matrix of explanatory variables. This revealed the following nontrivial multicollinearities: (1) $DHW_{90}$ with $CSA_{During}$; (2) $DTR_{90}$ with $DTR_{30}$; (3) MMM with $MMM_{Total}$; and (4) $DTR_{Total}$ with $DTR_{SS}$ and $DTR_{FW}$ (Table 1). Note that within each of these four multicollinear groupings, the multicollinear variables come from the same explanatory variable category (for example, both $DHW_{90}$ and $CSA_{During}$ are both from the Cumulative Thermal Stress category). Therefore, for each logit model, we selected at most one variable from a category without multicollinearity erroneously influencing our results. Spatial autocorrelation within each covariate was determined by calculating Moran's $I$ and examining correlograms, from which we determined significant spatial autocorrelation typically up to distances of 500 km (Supplementary Fig. 5). To account for this, we also added a random effect to the highest-ranked logit models, from which we failed to conclude a marginal model improvement (see below).

All permutations of all possible explanatory variables were used to compute a total of 10,367 logit models, where all logit models were computed using a multinomial logistic regression function ("mnrfit") in MATLAB. Model comparison was performed using a bias correction for small sample sizes to Akaike's Information Criterion, $AIC_C$[80], and all models within $\Delta AIC_C \le 2$ of the best model ($\Delta AIC_C = 0$), which have statistically indistinguishable performances[81], are presented in Supplementary Fig. 6. McFadden's pseudo-$R^2$ was also computed for the highest ranked models, and ranged from 0.26 to 0.30, with that of the highest-ranked model equal to 0.30. While the logit model with the lowest $AIC_C$, as well as all models within $\Delta AIC_C \le 2$, provide a general outline for the coefficients of the critical independent variables in explaining bleaching prevalence, these parameter estimates have errors of unknown distribution. Additionally, the possible existence of outliers in the high-frequency temperature variability data may influence the results of the logit parameter estimates, and therefore, delete-1 jackknife resampling was used to compute estimates of bias and standard errors. We found a slight positive bias that did not significantly alter the influence of high-frequency temperature variability relative to the other covariates. Estimates of standard errors for all logit parameters of all models within $\Delta AIC_C \le 2$ can be seen in standard error bars (Fig. 3b and Supplementary Fig. 6). Specifically, the error bars seen in Fig. 3c are the standard errors from each contribution summed in quadrature. A modified jackknife resampling scheme was also performed, in which instead of leaving out one site at a time, sites within 10 km of each other were grouped together and each of these proximity groups was left out incrementally before fitting OLR models to the remaining data. This spatial resampling analysis (Supplementary Fig. 7) did not result in significantly different parameter estimates than the full model presented in Fig. 3b. Estimates for the intercept terms $C_1$, $C_2$, and $C_3$ (±non-resampled standard errors) were found to be $0.72 \pm 0.35$, $1.41 \pm 0.38$, and $2.00 \pm 0.42$, respectively, indicating no significant difference between bleaching prevalence categories within our dataset. Using the statistical computing software R[82] (https://www.r-project.org), a random effect grouping variable was added to each of the highest-ranked models, which grouped reefs within 5 km of each other. These resulting mixed effects models were compared to their fixed effects equivalents to determine model fit improvement, and the inclusion of a random effect did not improve the fit of any of the 20 highest-ranked models (Supplementary Data 3). Furthermore, to account for possible nonlinear interactions between covariates, such as $Depth \times DTR_{30}$ or $Acute1 \times DHW_{30}$, we also included a nonlinear interaction term to the highest-ranked model and determined model fit improvement. The $AIC_C$ of these nonlinear models (Supplementary Table 1) indicated that the addition of a nonlinear interaction term did not significantly improve the fit of the original highest-ranked model displayed in Fig. 3b. Therefore, as our main result, we ultimately report the fixed effects ORL model parameter estimates with no nonlinear or interaction terms (Fig. 3b).

The SST-based OLR analysis summarized in Fig. 5 was performed using the CoRTAD 4 km weekly SST data[55] pixels that were closest to our in situ bleaching observations. The quantities MMM, $MMM_{Max}$[15], Acute1, Acute2, DHW, $ROTC_{SS}$, and $ROTC_{90}$ (Table 1) were computed from the CoRTAD temperature data. This resulted in a total of 27 covariates (20 as described above, and 7 new ones computed from the CoRTAD data). We then proceeded to fit OLR models using all

permutations of covariates, with the constraint that within each model, we only include ≤1 covariate from each category listed in Table 1. This produced 60,479 models, from which we performed model comparison using $AIC_C$, resulting in 12 dominant models being identified as statistically indistinguishable. From here, we concluded that when 4 km SST metrics are incorporated into our model framework, high-frequency temperature variability and depth remain the most influential covariates on bleaching prevalence, acting to attenuate bleaching (Fig. 5). Furthermore, acute and cumulative thermal stress exacerbate bleaching prevalence the most, and the SST-derived version of the latter contributes substantially to this effect. The results presented in Fig. 5 ultimately mirror those from Fig. 3b, c, indicating that DTR is the main driver of bleaching prevalence, regardless of whether SST or in situ quantities of the other predictors are used. Common covariates in these models were MMM, $MMM_{Max}$, $MMM_{4\ km}$, $DHW_{30}$, $DHW_{4\ km}$, Acute1, $Acute2_{4\ km}$, and $ROTC_{90\text{-}4\ km}$, and each of these 12 highest-ranked models included at least one SST-based covariate. The $AIC_C$ values of these models ranged from 129.19 to 131.12, with McFadden's pseudo-$R^2$ values ranging from 0.35 to 0.38, which presents a considerable improvement to the highest-ranked model that was fit to only in situ values, which had an $AIC_C$ equal to 143.75 and a McFadden's pseudo-$R^2$ of 0.30. To highlight the importance of High-Frequency Temperature Variability in model improvement, we also fit these models to the data and excluded the Daily Temperature Range covariate, which was either $DTR_{30}$ or $DTR_{90}$. This significantly decreased model performance, with $AIC_C$ values ranging from 153.46 to 169.57, and McFadden's pseudo-$R^2$ values ranging from 0.12 to 0.23, which supports the idea that $DTR_{30}$ is the dominant driving variable for the bleaching response.

**Data availability**. All 118 temperature time series used in this study are archived with the NOAA National Centers for Environmental (NCEI), with accession number 0170826 [83].

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

## Acknowledgements

The authors thank the National Data Buoy Center and the Florida Institute of Oceanography for the use of data from the Florida Keys, and the Moorea Coral Reef Long-Term Ecological Research site for South Pacific data. The authors thank the UC Irvine Data Science Initiative, as well as the UCI OCEANS Initiative, for funding to support this work. The authors also thank Professor Peter J. Edmunds and Professor Steven J. Davis for their assistance in writing this manuscript. This is CSUN Marine Biology contribution #267. The authors acknowledge that A. Safaie and K. Davis were supported by National Science Foundation Award No. 1436254 and G. Pawlak was supported by Award No. 1436522.

## Author contributions

K.A.D. and A.S. designed the study. T.R.M., G.P., D.J.B., J.L.H., J.S.R., G.J.W., and K.A.D. collected the data. A.S., K.A.D., and N.J.S. analyzed the data. A.S., K.A.D., N.J.S wrote the paper, and all authors contributed significantly to the interpretation and editing of the manuscript.

## Additional information

**Competing interests:** The authors declare no competing interests.

