## [Peer Review File · Nature Communications]

Reviewers' comments:

Reviewer #1 (Remarks to the Author):

This manuscript aims to identify the main correlates of coral bleaching severity from a global dataset of 20 environmental variables (related to habitat, background conditions, and multiple metrics of thermal stress) at 118 reef locations where bleaching events were observed, concluding that daily thermal range was the main driver of bleaching prevalence. It is an impressive effort of data synthesis, analysis of thermal stress time series and identification of the main drivers of bleaching among ecologically meaningful but highly correlated variables. In this, I think it is a powerful message that will be of interest to coral reef ecologists as well as the broad scientific audience. The MS is generally very clear and well written, and I only a few very minor comments on the presentation (see below). I have listed some major comments on the analysis below, and I hope they will help strengthen this manuscript that –once revised– should make a convincing and novel contribution to the coral reef literature.

My main concerns lie with the main statistical analysis that leads to the identification of the DTR as the main driver of bleaching risk, i.e. the ordinal logistic regression. As far as I understand it, the authors did not account for the spatial dependency among their statistical units (individual reef sites). Some sites are only a few kms apart (e.g. Nelly Bay seaward and shoreward habitats), implying a hierarchical (nested) structure in the dataset at the site, reef location and reef region levels. This, in my opinion, poses the issue of pseudo replication whereby the observations are not independent (yet considered as such), which inflates the chance of Type-I errors. One way to account for this is to have a random effect (i.e. in a logistic mixed-effect regression) accounting for the hierarchical nature of the data, or some form of autoregressive model accounting for this spatial structure. Spatial autocorrelation in the raw data and the model residuals should also be assessed (e.g. using Moran's I correlograms).

I was also surprised that non-linear effects and possible interactions among predictors were not considered. An obvious case that comes to mind is the possibility of an interaction between depth and thermal stress – i.e. is the effect of DTR the same at shallow or deep-water reefs. Or is there a given depth or DTR that maximizes bleaching risk – which would imply possible non-linear effects e.g. bell shaped response curve, which could be tested using polynomial terms in the model.

On a more general note, it would be good to get an understanding of the variance around the model estimates and goodness-of-fit. For example, when the authors say that “a 1 °C increase in high-frequency temperature variability would decrease the probability of severe bleaching by 31%”, what is the variance of this effect and how does it vary spatially e.g. among reef regions? In Figure 2 (see further comments on this Fig below), there does not seem to be a great difference between the bleaching prevalence categories- I am wondering whether the corresponding error bars would be overlapping? Last, I realize the authors have already put in a great effort of resampling while selecting model predictors, but I am wondering how much the results would change if one of the region or some of sites were left out of the calibration dataset, which could be tested using some form of bootstrapping.

Finally, I find the presentation of the sample sizes confusing. I thought each bleaching event at a given location was associated with a given time series, so why 118 temperature time series for 81 bleaching events? Was that because multiple sites were used for the same 'bleaching event', and assigned the same temperature data? This would make the issue of pseudo-replication I mentioned above even more problematic... This aspect needs to be clarified early on, and the terminology between sites/locations/regions kept consistent throughout the MS.

MINOR COMMENTS

L31 Change 'limited' to 'coarse'

L34 Not only across, but also within reefs.

L82 Change 'which' to 'that'

L123 How were the six 'representative' time series selected, i.e. based on which criteria?

L152 Subtitle style needs to be consistent. Other subtitles refer to a theme, this one is about a specific result.

L243-245 But see e.g. Mellin et al. Ecology Letters 2016.

Figure 1. Showing the location of Nelly Bay and Tahala on the map would be helpful.

Figure 2. I find this figure difficult to interpret. First, I don't understand the first part of the y-axis label ($P(y>j)$). Also, as a probability it would make more sense to have it on a $[0,1]$ interval.

Regarding the x-axes, I don't understand why all covariates were not standardized to the same interval (e.g. 0,1 or -1,1) and it might help the interpretation if they were represented on their original scales.

Reviewer #2 (Remarks to the Author):

This study documents how an array of temperature-related variables correlates with coral bleaching, using a global suite of in-situ ocean temperature data from reef sites. The study adds to substantial previous work showing that a) temperature is a dominant variable controlling coral bleaching, and b) that corals previously exposed to variable temperatures on daily and shorter time scales tend to bleach less, compared to those growing in thermally uniform environments. The current study is novel in that it uses locally derived temperature data, rather than gridded/remotely sensed SST, and it explores various aspects of the temperature history, including rates of change, particular trajectories, higher order statistical properties, and high-frequency changes. They find a particularly important role for temperature variability on the scale of days or less. To my knowledge, this is a new (and important) result, made possible by the global array of in-situ reef temperature data that they have compiled.

The result is important because as coral reefs exhibit mass bleaching and mortality in a warming world, efforts to prioritize sites for conservation, predict and monitor bleaching events, identify mechanisms of heat resistance in corals, selectively breed "hardened" corals, and generally gain insight into how bleaching works can all be informed by the results of this work.

The dataset they use is obviously critical to the analysis, so more care should be taken to describe how it was assembled, what choices were made in searching for and selecting sites, etc. I'd like to see this dataset made public, with metadata, more proactively than "will provide when requested by the Editor". Archiving this through an appropriate NOAA data center would be a better alternative.

One analytical concern: They perform spectral analysis on their temperature data to identify significant periodic components of the variability. The presence of a strong diurnal cycle is no surprise from a physical standpoint, and a semidiurnal cycle may be reasonable in areas with semidiurnal tides. But the presence of "terdiurnal and quarterdiurnal" cycles raises a red flag – it's unclear what physical forcing might account for this. Their explanation is unsatisfying – I wouldn't expect winds and waves to generate clear periodicities of 3 and 4 per day. I'm left wondering whether there is a "ringing" in the spectral analysis from the dominant daily cycle that is artificially propagated into higher frequencies – in other words, that these higher frequency peaks are artifacts of the method. This point is not a critical one for their overall analysis, as the diurnal cycle (and in some places semidiurnal) is sufficient to support their conclusions. But they should explore this issue using alternate methods.

Another question that should be addressed is whether there are species-level differences between the high- and low-variability sites, and whether that accounts for the difference in bleaching. One might imagine their results could be explained by differences in dominant species between high and low thermal variability sites.

The choice of case studies illuminates the results with specificity, but its not clear why these sites were chosen. Are there many sites where such a comparison could be made? If so, what is the overall message from all these sites? How typical are Tahala and Nelly Bay?

More specific comments:

In the Table: how is "summation of temperature" calculated? Are the binary values in group 4 indicating a single incidence of high T? Over what time persistence?

In figure 1, caption (bold) – Principal not principle. Map could be blown up to highlight area of interest, making it more readable, by eliminating higher latitudes.

In figure 4: is the darker colored region indicating $j=4$?

Throughout – overuse of hyphens. ("remotely-sensed," "widely-used," "experimentally-heated", "in-situ:" these do not require hyphens, nor do any where the first word ends in -ly)

Reviewer #3 (Remarks to the Author):

This study investigates the role of high-frequency temperature variability on predictions of coral bleaching over small spatial and temporal scales (<1 km, daily). The major finding is that high frequency changes in temperature over daily timescales are associated with greater bleaching tolerance. While this finding itself is not novel (cited in the text; lines 43-56), the study makes a solid contribution to field by rigorously comparing in situ temperature data to bleaching records in 5 major reef regions around the world. For these reasons, I believe this study represents an important contribution to the coral reef community and the wider field. I also strongly encourage the authors to make their valuable raw data set easily and openly accessible.

One valuable aspect of a study like this is the potential to compare fine- and coarse scale temporal and spatial temperature variability in predicting bleaching. While the study does a nice job of isolating the Daily Temperature Range (DTR) as a major predictor at the reef scale (<1km, daily), I think there is a missed opportunity here to compare in situ temperature metrics with satellite 4km biweekly SST data (which is globally available) for bleaching predictions which would be more useful from a management standpoint. It is not novel to show that more thermally variable sites have more thermally tolerant corals, but this analysis could be easily extended using the same model framework to provide a valuable comparison across scales. This is my major recommendation.

Lines 43-62. It might be worth noting that other modelling studies at the global scale have also shown that including coarser scale temperature variability in the calculation of the SST climatology improves bleaching predictions:

Boylan, P., & Kleypas, J. (2008). New insights into the exposure and sensitivity of coral reefs to ocean warming. In Proc 11th Int Coral Reef Symp (Vol. 2, pp. 854-858).

Donner, S. D. (2011). An evaluation of the effect of recent temperature variability on the prediction of coral bleaching events. *Ecological Applications*, 21(5), 1718-1730.

Logan, C. A., Dunne, J. P., Eakin, C. M., & Donner, S. D. (2012, July). A framework for comparing coral bleaching thresholds. In *Proceedings of the 12th International Coral Reef Symposium* (ed Yellowlees D, Hughes TP), pp 10A3 Townsville [Internet].

Lines 74-105. I suggest breaking this paragraph up to make it more read-able. Perhaps you could discuss the categories that include a measure of high frequency variability in a 2nd paragraph, and the PCA analysis in a 3rd. Also I am only seeing 7/8 categories listed in this paragraph. I think you've accidentally omitted 'Acute thermal stress'.

Line 76-77. Is reef type included as a variable (e.g. fore vs back reef)? If not, could you comment on this?

Line 79-81, Lines 84-85, Lines 90-92. It would be a great contribution if the authors could include the MMM climatology (and perhaps MMMmax; Donner et al. 2011) calculated based on 4km satellite SSTs in their model to compare how remotely sensed data compares with the in situ data predictors. You do not necessarily need the NOAA CRW virtual stations for this; you can obtain the data from CORTAD (<https://data.noaa.gov/dataset/the-coral-reef-temperature-anomaly-database-cortad-version-5-global-4-km-sea-surface-temperatu07731>). This could be especially useful for locations that do not have in situ data available. This comment also applies to the categories of "Background Conditions", "Cumulative Thermal Stress" and "Rate of Heating". The kind of analysis performed here could also be very useful for improving bleaching predictions at coarser spatial scales especially because among the highest-ranking logit models, cumulative (and acute) thermal stress largely contributed to bleaching (Lines 188-190

Lines 221. Omit "within coral genes" as it is not necessary.

Line 227. Define what is meant by "encourage". Can you comment on what specific mechanism(s) you are referring to (acclimatization, community shifts, adaptation)?

Lines 228-230. Globally, it is hard to disentangle if similar bleaching patterns across regions are a result of community shifts towards more tolerant species over time vs acclimatization or natural selection. I think the discussion warrants further discussion of how which mechanisms could lead the observed findings across regions.

Table 1. Under the Category 5 row, 'stress' is mis-spelled.

Table 1. Perhaps re-order the categories into those that include a measure of "high frequency variability" (e.g., 4, 7, 8) vs those that do not (1,2, 3?, 5). I had a hard time teasing apart which metrics required daily temperatures to calculate (vs those that could potentially be calculated using satellite SSTs).

Point-by-point response to comments on “High-frequency temperature variability reduces the risk of coral bleaching.”

September 2017

Reviewers' comments in black; Author responses in blue text. Changes to the manuscript have been highlighted in the revised version.

Reviewers' comments:

Reviewer #1 (Remarks to the Author):

This manuscript aims to identify the main correlates of coral bleaching severity from a global dataset of 20 environmental variables (related to habitat, background conditions, and multiple metrics of thermal stress) at 118 reef locations where bleaching events were observed, concluding that daily thermal range was the main driver of bleaching prevalence. It is an impressive effort of data synthesis, analysis of thermal stress time series and identification of the main drivers of bleaching among ecologically meaningful but highly correlated variables. In this, I think it is a powerful message that will be of interest to coral reef ecologists as well as the broad scientific audience. The MS is generally very clear and well written, and I only a few very minor comments on the presentation (see below). I have listed some major comments on the analysis below, and I hope they will help strengthen this manuscript that –once revised– should make a convincing and novel contribution to the coral reef literature.

My main concerns lie with the main statistical analysis that leads to the identification of the DTR as the main driver of bleaching risk, i.e. the ordinal logistic regression. As far as I understand it, the authors did not account for the spatial dependency among their statistical units (individual reef sites). Some sites are only a few kms apart (e.g. Nelly Bay seaward and shoreward habitats), implying a hierarchical (nested) structure in the dataset at the site, reef location and reef region levels. This, in my opinion, poses the issue of pseudo replication whereby the observations are not independent (yet considered as such), which inflates the chance of Type-I errors. One way to account for this is to have a random effect (i.e. in a logistic mixed-effect regression) accounting for the hierarchical nature of the data, or some form of autoregressive model accounting for this spatial structure. Spatial autocorrelation in the raw data and the model residuals should also be assessed (e.g. using Moran's I correlograms).

Reviewer #1 identifies an issue we did not account for in our ordinal logistic regression (OLR) models, which is the degree of spatial autocorrelation amongst samples. While two habitats within the same reef are indeed likely to be correlated to some degree (i.e. low frequency variations), we tried to emphasize in the main text that on very finely resolved spatiotemporal scales (which is at the spirit of this study), hydrodynamics and temperature variability at two habitats within a reef can be rather different (Lines 143-158, 236-248) when considering variance and diurnal temperature ranges. The primary example we used to illustrate this point is illustrated by Figure 6, which depicts the different thermal variability environments at reefs in Tahala and Nelly Bay. However, as Reviewer #1 correctly instructs, a more rigorous analysis of the correlation between thermal microenvironments is warranted in order to fulfill assumptions

of independence required for subsequent statistical tests. To address this, we computed Moran's I and plotted correlograms for all 20 predictors used in the ordinal logistic regression, as well as the residuals of the highest-ranked models (using 1-tailed assumptions of positive spatial autocorrelation). From these correlograms, especially for that of DTR₃₀, which we will add to the Supplementary Info as Supplementary Figure 5, (also shown below), we concluded significant spatial autocorrelation up to distances of 500 km. We then re-fit the 20 highest-ranked OLR models (all models with $\Delta AIC_C \leq 2$) with the addition of a random effect. We did this twice; the first random effect we added to all 20 models was to account for reef region (Pacific, Caribbean, Western Indian Ocean, Red Sea, Great Barrier Reef), and the next random effect accounted for reef distances (proximities) to each other, in which we grouped reefs that were within 5 km of each other. It is this latter mixed effects modeling scheme that we will report on now, with results summarized in the attached Excel table ('Supplementary_Table_3.xlsx', which will also be included in the Supplementary Information as Supplementary Table 3). We initially note that the interpretation of the covariates remains the same even when a random effect accounting for reef proximities is added: Depth and DTR₃₀ remain bleaching 'attenuators', with DTR₃₀ remaining the most influential covariate of all, while the other covariates serve to exacerbate bleaching. Furthermore, from the model AIC_C values between fixed and mixed effects model pairs, the addition of a random effect does not significantly improve model fits. Likelihood ratio tests and McFadden's pseudo-R² values also enable us to make the same conclusion. While we ultimately acknowledge that accounting for spatial autocorrelation amongst our observations by including a random effect in the models adds to the robustness of our framework, we conclude that reporting the results of the fixed effects-only models in the Main Text of our study is sufficient. Furthermore, as we are striving to balance model performance with model parsimony (as is generally the case in ecology), there is no justifiable improvement in model performance to warrant increasing model complexity in this manner. Therefore, we will make the following additions to the Main Text and Supplementary Information:

- To the **Ordinal Logistic Regression** paragraph of *Methods*, we've added the following passage: "Spatial autocorrelation within each covariate was determined by calculating Moran's I and examining correlograms, from which we determined significant spatial autocorrelation typically up to distances of 500 km (Supplementary Figure 5). To account for this, we also added a random effect to the highest-ranked logit models, from which we failed to conclude a marginal model improvement (see below). A random effect grouping variable was added to each of the highest-ranked models, which grouped reefs within 5 km of each other. These resulting mixed effects models were compared to their fixed effects equivalents to determine model fit improvement, and the inclusion of a random effect did not improve the fit of any of the 20 models (Supplementary Table 3)."
- To the Supplementary Info, the following figure and caption were added:

“**Supplementary Figure 5| Moran’s I Correlograms.** For the covariates in the highest-ranked OLR model, Moran’s I was computed using distance classes of 1: [0, 0.5); 2: [0.5, 5.0); 3: [5.0, 50.0); 4: [50.0, 500.0); 5: [500.0, 6145.0); 6: [6145.0, 6400.0); 7: [6400.0, 10000.0); 8: [10000.0, 13000.0); 9: [13000.0, ∞). 95% confidence intervals are shown, and critical values for Moran’s I were computed using a correction for small sample sizes. Covariates in the highest-ranked *in situ* model include a) Depth, b) DHW₃₀, c) Acute1, d) Thermal Trajectory, e) ROTC_{SS}, and f) DTR₃₀.”

- An Excel table (‘Supplementary_Table_3.xlsx’) was added to the Supplementary Info detailing the random effects models and their fits, along with the following caption:

“**Supplementary Table 3| Results of Mixed Effects Ordinal Logistic Regression Models.** After identifying the covariates and their coefficients from the highest-ranked models ($\Delta AIC \leq 2$), we computed these same models with the addition of a random effect accounting for reef proximities to each other. Reefs within 5 km of each other were grouped together. Parameter estimates for the intercepts, as well as the covariate coefficients included in the each of the highest-ranked models, are shown below. Also given are the standard deviations for the random effects terms in the mixed effects models, AICC values, likelihood ratio statistics and the associated *p*-values, and McFadden’s pseudo R² values. Mixed effects models were computed using the equation below, and therefore in this case, covariates with negative coefficients act as bleaching ‘mitigators.’”

I was also surprised that non-linear effects and possible interactions among predictors were not considered. An obvious case that comes to mind is the possibility of an interaction between depth and thermal stress – i.e. is the effect of DTR the same at shallow or deep-water reefs. Or is there a given depth or DTR that maximizes bleaching risk – which would imply possible non-linear effects e.g. bell shaped response curve, which could be tested using polynomial terms in the model.

Reviewer #1 makes an important suggestion here by calling to attention nonlinear effects in the OLR model. Our initial effort to address the interpretation of depth and DTR in the highest-

ranked OLR models was to cite publications (Lines 182-194) demonstrating how depth and high-frequency temperature variability can mitigate thermal stress and bleaching in complementary ways. However, in response to Reviewer #1's comment, in short, we did not see evidence for non-linearity (*i.e.* polynomial terms), and we proceeded to test for interactions amongst the covariates, which ultimately failed to yield significant model improvement. We therefore fitted new nonlinear OLR models, adding in an interaction term and comparing the AIC_C of these nonlinear models with that of the originally reported highest-ranked model (bleaching \sim Depth + DHW_{30} + Acute1 + TT + ROTC + DTR_{30}). The interactions we included were chosen either because of their physically-meaningful relationship (*i.e.* Depth x Daily Temperature Range), or because of the way in which the covariates were computed (*i.e.* MMM x DHW_{30} or Acute1 x DHW_{30}). Supplementary Table 4 summarizes the interaction terms we used, the resulting coefficients for the thermal stress and high-frequency temperature variability covariates (DHW_{30} and DTR_{30} , respectively), the coefficient for the interaction (nonlinear) term, and the model AIC_C value. In summary, while some interactions had notable effects, the strongest of which were from Acute1 x DHW_{30} (coefficient = -3.94), Depth x DTR_{30} (coefficient = 2.69), and Depth x DHW_{30} (coefficient = 1.78), the inclusion of an extra interaction term did not significantly improve model fits. The relatively large coefficient for the Acute1 x DHW_{30} term makes intuitive sense in that Acute1 measured whether temperatures exceeded 1°C over the maximum monthly mean climatology, and this threshold is exactly the baseline over which DHW_{30} is calculated. The interactions of Depth with DTR_{30} and DHW_{30} also seem physically intuitive, and essentially comprise the thesis of this study. The following revisions were made to the Main Text and Supplementary Information:

- To the **Ordinal Logistic Regression** paragraph of the *Methods* section in the Main Text, we've added the following lines: "Furthermore, to account for possible nonlinear interactions between covariates, such as depth x DTR_{30} or Acute1 x DHW_{30} , we also included a nonlinear interaction term to the highest-ranked model and determined model fit improvement. The AIC_C of these nonlinear models (Supplementary Table 4) indicated that the addition of a nonlinear interaction term did not significantly improve the fit of the original highest-ranked model displayed in Figure 3b." This addition has also been highlighted.
- To the Supplementary Information, we've added the following table and caption:

Interaction	Interaction Coefficient	Coefficient for:		AICc
		DHW ₃₀	DTR ₃₀	
depth x DHW ₃₀	1.78	2.11	-2.31	143.57
depth x DTR ₃₀	2.69	1.77	-1.72	142.74
depth x ROTC	-0.09	0.95	-2.63	146.32
depth x MMM	-0.11	0.96	-2.6	146.31
MMM x DHW ₃₀	0.8	2.04	-2.85	144.41
MMM x TT	-0.4	0.83	-2.89	145
MMM x Acute1	0.32	1.14	-2.8	145.3
MMM x ROTC	-0.01	0.97	-2.67	146.35
Acute1 x DHW ₃₀	-3.94	5.99	-3.06	143.37
TT x DHW ₃₀	-0.55	2.1	-2.51	144.27
ROTC x DTR ₃₀	-0.15	0.97	-2.63	146.32

Supplementary Table 4: Nonlinear Ordinal Logistic Regression Models. Coefficients for *in situ* interaction terms that were added to the highest-ranked OLR model, as well as the coefficients for the DHW₃₀ and DTR₃₀ covariates, and the new model AIC_C are shown. The AIC_C for the original highest-ranked model was 143.75.

On a more general note, it would be good to get an understanding of the variance around the model estimates and goodness-of-fit. For example, when the authors say that “a 1 °C increase in high-frequency temperature variability would decrease the probability of severe bleaching by 31%”, what is the variance of this effect and how does it vary spatially e.g. among reef regions? In Figure 4 (see further comments on this Fig below), there does not seem to be a great

difference between the bleaching prevalence categories- I am wondering whether the corresponding error bars would be overlapping? Last, I realize the authors have already put in a great effort of resampling while selecting model predictors, but I am wondering how much the results would change if one of the region or some of sites were left out of the calibration dataset, which could be tested using some form of bootstrapping.

To estimate the explained variability as well as the improvement from the null (intercept only) to fitted models, we computed McFadden's pseudo- R^2 for the OLR models. Here standard ordinary least squares (OLS) R^2 values cannot be computed, as the response variable is not continuous. For the highest-ranked models, McFadden's pseudo- R^2 ranged from 0.26 to 0.30. We initially did not report our pseudo R^2 values in the text, because pseudo- R^2 values, while still ranging from 0 to 1, are typically low, and this may unnecessarily arouse suspicion in audiences accustomed to OLS R^2 values (Hosmer Jr, D. W., Lemeshow, S., & Sturdivant, R. X. (2013). *Applied logistic regression* (Vol. 398). John Wiley & Sons.).

- The following addition has been made to the **Ordinal Logistic Regression** paragraph of the *Methods* section of the Main Text: "McFadden's pseudo- R^2 was also computed for the highest ranked models, and ranged from 0.26 to 0.30, with that of the highest-ranked model equal to 0.30."

Delete-1 jackknife resampled standard error bars for each of the parameters in the highest-ranked model are shown in Figure 3b of the Main Text.

There is little difference in the bleaching prevalence categories because they only differ by an intercept; this was fundamental to the assumption of proportional odds (parallel regression).

These intercepts were not shown in the previous figures, but the intercepts (\pm non-jackknifed s.e.) distinguishing one bleaching category from the next are $C_1 = 0.72 \pm 0.35$, $C_2 = 1.41 \pm 0.38$, and $C_3 = 2.00 \pm 0.42$ (shown below).

- To the **Ordinal Logistic Regression** section of the *Methods* section of the Main Text, we've added the following: "Estimates for the intercept terms C_1 , C_2 , and C_3 (\pm non-resampled standard errors) were found to be 0.72 ± 0.35 , 1.41 ± 0.38 , and $C_3 = 2.00 \pm 0.42$, respectively, indicating no significant difference between bleaching prevalence categories within our dataset."

Therefore, the estimates of the intercepts separating one bleaching threshold from another do indeed overlap. Ultimately this hints at a limitation brought about by an irregular sampling distribution in our response variable data, whereby 57% of the observations come from bleaching category 1, 16% from categories 2 and 3, and 27% from category 4.

Regarding resampling: we cannot leave out entire regions, because a great proportion of the covariate data comes from one region (GBR), and leaving that region out would be problematic for the iterative solver we're using to fit the OLR model. Also, as Reviewer #1 suggests we leave out entire reef regions or spatial groups, we interpret this as more of a delete-n jackknife resampling analysis as opposed to a straightforward bootstrapping technique. Therefore, we grouped sites based on their proximities to each other (sites within 10 km to each other were placed in the same group), similar to how we approached the issue of spatial autocorrelation above. We then proceeded to run a modified jackknife resampling analysis whereby instead of leaving one site out at a time, we left out one proximity group out at a time, such that on average 4 sites were left out at a time. We first compared the parameter estimates from the highest-ranked model fit to the entire dataset with the average of those from the resampled fits (see Supplementary Figure 7, also shown below). Ultimately, the parameter estimates for the full model were very similar to those from the resampling scheme; for example, the full model coefficients for Depth, DHW₃₀, and DTR₃₀ were 1.85, -0.96, and 2.66, respectively, while the average values from the resampled fits were 1.88, -1.05, and 2.71, respectively. Additionally, standard error bars between the full model and the resampled estimates overlap, indicating that we cannot conclude a significant difference in our highest-ranked model parameter estimates from this resampling scheme. The following additions were made to the Main Text and Supplementary Information:

- We've added the following lines to the **Ordinal Logistic Regression** paragraph of the *Methods* section of the Main Text: "A modified jackknife resampling scheme was also performed, in which instead of leaving out one site at a time, sites within 10 km of each other were grouped together and each of these proximity groups was left out incrementally before fitting OLR models to the remaining data. This spatial resampling analysis (Supplementary Figure 7) did not result in significantly different parameter estimates than the full model presented in Figure 3b".

- The following figure and caption was added to the Supplementary Information, and this change was highlighted:

Supplementary Figure 7| Highest Ranked *In Situ* Model Results After Spatial Resampling.

Standardized coefficients for the highest ranked model parameter estimates fit to the entire data set are shown in white (\pm s.e.), while the average of the resampled parameter estimates for the same covariates are shown in grey (\pm s.e.). The modified jackknife resampling scheme involved grouping sites within 10 km of each other, and then removing each of these proximity groups incrementally before fitting OLR models to the remaining data.

Finally, I find the presentation of the sample sizes confusing. I thought each bleaching event at a given location was associated with a given time series, so why 118 temperature time series for 81 bleaching events? Was that because multiple sites were used for the same ‘bleaching event’, and assigned the same temperature data? This would make the issue of pseudo-replication I mentioned above even more problematic... This aspect needs to be clarified early on, and the terminology between sites/locations/regions kept consistent throughout the MS.

Reviewer #1 brings up an important point. Not all of the *in situ* temperature time series were used in the bleaching analysis. The 81 bleaching events are instances where a survey (aerial or underwater) was done and the severity or prevalence of bleaching was recorded, and not all of the 118 time series were paired with a corresponding bleaching event. Multiple sites/time series were not used for the same bleaching observation; rather, for a given bleaching observation with its own unique time and location, we matched the closest of our temperature time series to that observation, if it overlapped the bleaching observation in time and was on the same relative habitat within the reef. The only way a temperature time series was repeatedly used in the bleaching analysis was if it had a corresponding bleaching observation occurring at two separate times, and even then, metrics and statistics computed from that temperature time series would

still be different because the vast majority of these metrics were time-dependent (i.e. ‘30 days preceding a bleaching observation’ etc). We made the following changes to the Main Text:

- The words ‘a pool of’ was included in Line 68.
- The word ‘locations’ was replaced with ‘habitats’ in Line 70.
- The sentence ‘Each of the 81 bleaching observations was matched to its own spatiotemporally coincident temperature time series data, such that not all of the 118 temperature time series were used in the subsequent bleaching analysis.’ was added to the introductory paragraphs of the Main Text (Lines 72-74).

MINOR COMMENTS

L31 Change ‘limited’ to ‘coarse’

The change was made.

L34 Not only across, but also within reefs.

The words “and within” were added in that line.

L82 Change ‘which’ to ‘that’

The change was made.

L123 How were the six ‘representative’ time series selected, i.e. based on which criteria?

These time series were chosen because there was another time series available within the same reef but at a different, relatively nearby, habitat. The time series whose PSDs are shown in Figure 2a are just the shallower or more shoreward of each pair.

L152 Subtitle style needs to be consistent. Other subtitles refer to a theme, this one is about a specific result.

The subtitle was changed to “The effect of diurnal temperature variability on bleaching”.

L243-245 But see e.g. Mellin et al. Ecology Letters 2016.

The reference was cited, and the first portion of that paragraph was changed to:

“Urgent global efforts at reducing anthropogenic greenhouse gas emissions must remain a priority for reef preservation, due to the acute thermal stress that may arise from global warming projections on reefs^{5,57}. However, combating the effects of local stressors on reefs through conservation tools such as marine protected areas (MPAs) is likely to increase the chances of reef persistence through future warming, as well as to facilitate recovery following a bleaching event^{2,50,58,59}. Warming ocean temperatures...”

Figure 1. Showing the location of Nelly Bay and Tahala on the map would be helpful.

The text was added to the map to indicate the locations of Nelly Bay and Tahala.

Figure 2. I find this figure difficult to interpret. First, I don’t understand the first part of the y-axis label ($P(y>j)$). Also, as a probability it would make more sense to have it on a $[0,1]$ interval. Regarding the x-axes, I don’t understand why all covariates were not standardized to the same interval (e.g. 0,1 or -1,1) and it might help the interpretation if they were represented on their original scales.

The y-axis label was changed to ‘Probability’, as these plots show the pdfs of observing a bleaching prevalence category. The y-axis ticks were changed to a $[0,1]$ interval. The x-axis ticks and label will remain the same, because we would like to have them in their native units to make the scale more interpretable (i.e. an increase of 10 m in depth would dramatically reduce the probability of any bleaching category).

Reviewer #2 (Remarks to the Author):

This study documents how an array of temperature-related variables correlates with coral bleaching, using a global suite of in-situ ocean temperature data from reef sites. The study adds to substantial previous work showing that a) temperature is a dominant variable controlling coral bleaching, and b) that corals previously exposed to variable temperatures on daily and shorter time scales tend to bleach less, compared to those growing in thermally uniform environments. The current study is novel in that it uses locally derived temperature data, rather than gridded/remotely sensed SST, and it explores various aspects of the temperature history, including rates of change, particular trajectories, higher order statistical properties, and high-frequency changes. They find a particularly important role for temperature variability on the scale of days or less. To my knowledge, this is a new (and important) result, made possible by the global array of in-situ reef temperature data that they have compiled.

The result is important because as coral reefs exhibit mass bleaching and mortality in a warming world, efforts to prioritize sites for conservation, predict and monitor bleaching events, identify mechanisms of heat resistance in corals, selectively breed “hardened” corals, and generally gain insight into how bleaching works can all be informed by the results of this work.

The dataset they use is obviously critical to the analysis, so more care should be taken to describe how it was assembled, what choices were made in searching for and selecting sites, etc. I’d like to see this dataset made public, with metadata, more proactively than “will provide when requested by the Editor”. Archiving this through an appropriate NOAA data center would be a better alternative.

We are thankful to Reviewer #2 for the encouraging comments and their thoughtful criticism.

- Regarding the data assembly, the following text was added to the first paragraph in the *Methods* section (Data Synthesis): “Many of these time series were obtained directly from the researchers (Supplementary Table 1), or from publicly-available databases including Australian Institute of Marine Science (AIMS), the National Data Buoy Center (NDBC), and the Florida Institute of Oceanography (FIO). However, time series within our dataset were also selected to match precise reef locations that had sufficiently documented bleaching events, while also containing as long and consistently sampled records as possible.”

We’ve also included the following Data Availability statement in our Manuscript:

Data Availability

No restriction; the authors will upload the data to the Biological and Chemical Oceanography Data Management Office (BCO-DMO) and the NOAA National Centers for Environmental Information (NCEI) immediately upon (approval for) publication.

One analytical concern: They perform spectral analysis on their temperature data to identify significant periodic components of the variability. The presence of a strong diurnal cycle is no surprise from a physical standpoint, and a semidiurnal cycle may be reasonable in areas with semidiurnal tides. But the presence of “terdiurnal and quarterdiurnal” cycles raises a red flag – it’s unclear what physical forcing might account for this. Their explanation is unsatisfying – I

wouldn't expect winds and waves to generate clear periodicities of 3 and 4 per day. I'm left wondering whether there is a "ringing" in the spectral analysis from the dominant daily cycle that is artificially propagated into higher frequencies – in other words, that these higher frequency peaks are artifacts of the method. This point is not a critical one for their overall analysis, as the diurnal cycle (and in some places semidiurnal) is sufficient to support their conclusions. But they should explore this issue using alternate methods.

We are grateful to Reviewer #2 for reminding us to speak more on this issue. We certainly agree that the ter- and quarterdiurnal periodicities, whether marked as significant or not, are more than likely artifacts from our spectral analysis method (largely based on the `pwelch()` command from MATLAB).

- Therefore, we have changed and highlighted the text in the first sentence of the first paragraph of the Results section (Spatiotemporal dependence of diurnal temperature variability, Lines 118-120) to: “The thermal metrics computed from temperature time series were highly variable across sites, but regardless of location and depth, all 118 time series show significant temperature variations in the high-frequency band, which we define as 0.727-4 cpd (Supplementary Fig. 1), though it may be that the spectral peaks occurring on 3-4 cpd frequencies are harmonics of the diurnal and semidiurnal peaks.”

We recomputed our spectral peaks using different window functions (Hamming or Flat Top), interpolated sampling intervals, window sizes, and number of discrete Fourier transform points used in the PSD estimate, and the results of these different computations still show significant spectral peaks at 3-4 cpd, as seen in the figure below. Therefore, we will only indicate that they are likely artificial harmonics, and we will not emphasize their importance in the main text.

In this figure, PSDs are computed for 6 different time series (rows); each column represents a different method of computing the PSD. Column a: 2 hour interpolated sampling interval with a

hamming window, and window sizes are 36 months (for time series longer than 10 years) or 4 months (for shorter time series). Column **b**: Default** sampling interval with a flat top window, and window sizes are 36 months (for time series longer than 10 years) or 4 months (for shorter time series). Column **c**: Default** sampling interval with a hamming window, and window sizes are 9 months (for time series longer than 10 years) or 2 months (for shorter time series). Column **d**: Default** sampling interval with a hamming window, and window sizes are 36 months (for time series longer than 10 years) or 4 months (for shorter time series), and number of discrete Fourier points is set such that 365 days is 1 period. Column **e**: Combination of columns **a** and **d**. ** *The default sampling interval is the most consistent sampling interval w/in a given time series. In most cases, this was clear to identify. In rare cases, such as when a logger was swapped out or replaced and a different sampling interval was used, then the larger of the two intervals was chosen.*

Another question that should be addressed is whether there are species-level differences between the high- and low-variability sites, and whether that accounts for the difference in bleaching. One might imagine their results could be explained by differences in dominant species between high and low thermal variability sites.

Reviewer #2 identifies what we believe to be the main weakness of our bleaching analysis. We initially hoped to be able to include species data in our analysis, however there is insufficient species-level information for a clear majority of the observations. Very rarely are we able to identify dominant species associated with a survey site, and even when we are, the same group of species are present at multiple sites within a given reef region so as to obscure the interpretation. Frankly, this is the major limitation of our study, and a valuable follow up would be to include species-level information within our model framework.

- To the **Discussion** section of the Main Text, we've added the following (Lines 263-267):
“While we lack sufficient species-level data, we fully acknowledge that intrinsic coral properties³⁹, differences in reef-scale community compositions⁵³, and taxonomic susceptibility²³ are likely to influence heterogeneous reef-scale bleaching responses, and an improvement to our model framework would include species-level covariates.”

The choice of case studies illuminates the results with specificity, but its not clear why these sites were chosen. Are there many sites where such a comparison could be made? If so, what is the overall message from all these sites? How typical are Tahala and Nelly Bay?

We are glad to hear that Reviewer #2 finds the case studies informative, and in short, these sites were largely chosen for their data availability. These sites were the only ones that allowed us to take an in depth look at instances of bleaching heterogeneity amongst locations within different habitats of the same reef, thus emphasizing the importance of high-frequency temperature variability in shaping the thermal tolerance of corals. Furthermore, we had supplementary data on the environmental/meteorological conditions at these reefs to help with the interpretation of their water temperature time series and bleaching events. The PCA (Figure 1) shows where the Tahala and Nelly Bay seaward sites (blue triangles) and shoreward sites (red squares) fall along the *in situ* covariates; Tahala seems to be exceptionally driven by heating rate and background conditions, whereas the Nelly Bay reef slope is more similar to the remaining sites, and the Nelly Bay reef flat is largely driven by cumulative heat stress. Like all the other temperature time series, these sites demonstrate strong diurnal variability (Supplementary Figure 1), and furthermore, the Nelly Bay reef flat and slope sites seem to have rather typical MMM, DTR, and

ROTC values (see figure below). The Tahala reef sites have relatively large MMMs compared to the remainder of the data, and the wave-protected site has high diurnal temperature variability, which is to be expected (Davis et al 2011). Ultimately, while the Tahala sites do exhibit strong heating behavior in general, we will allow the case study to remain as presented in the Main Text.

- To the **Specific within-reef cases** paragraph of the **Results** section of the Main Text, we've added and highlighted the following (Lines 241-243): “These sites were chosen due to the availability of supplementary data of the environmental and meteorological conditions^{29,53} at these reefs, which aided in the analyses of their water temperature time series and interpretations of their bleaching events.”

More specific comments:

In the Table: how is “summation of temperature” calculated? Are the binary values in group 4 indicating a single incidence of high T? Over what time persistence?

The calculations were done using trapezoidal integration (the trapz() function in MATLAB); “Summation of temperature” was changed to “Trapezoidal integration of temperatures” and highlighted in all instances in Table 1.

The binary values in group 4 indicate whether or not ANY daily mean temperatures within 90 days preceding a bleaching date exceeded MMM + 1 °C (Acute1) or MMM + 2 °C (Acute2); even a single incidence of high daily mean temperature could trigger a 1. In the Table, sentence was changed to “Binary value indicating whether any of the daily mean temperatures within 90 days preceding a bleaching event exceeded...”.

In figure 1, caption (bold) – Principal not principle. Map could be blown up to highlight area of interest, making it more readable, by eliminating higher latitudes.

The word principle was replaced with principal.

The latitudinal bounds on the map were changed to zoom in on the study sites.

In figure 4: is the darker colored region indicating $j=4$?

In Figure 4, only $j = 1$ to 3 are shown; $j=4$ is a reference category and can be interpreted as $1 - P_{j=3}$.

Throughout – overuse of hyphens. (“remotely-sensed,” “widely-used,” “experimentally-heated”, “in-situ.” these do not require hyphens, nor do any where the first word ends in -ly)

Hyphens were removed from these phrases and anywhere the first word ends in -ly.

Reviewer #3 (Remarks to the Author):

This study investigates the role of high-frequency temperature variability on predictions of coral bleaching over small spatial and temporal scales (<1 km, daily). The major finding is that high frequency changes in temperature over daily timescales are associated with greater bleaching tolerance. While this finding itself is not novel (cited in the text; lines 43-56), the study makes a solid contribution to field by rigorously comparing in situ temperature data to bleaching records in 5 major reef regions around the world. For these reasons, I believe this study represents an important contribution to the coral reef community and the wider field. I also strongly encourage the authors to make their valuable raw data set easily and openly accessible.

Reviewer #3 brings up an important point here. We’ve included the following Data Availability statement in our Manuscript:

Data Availability

No restriction; the authors will upload the data to the Biological and Chemical Oceanography Data Management Office (BCO-DMO) and the NOAA National Centers for Environmental Information (NCEI) immediately upon (approval for) publication.

One valuable aspect of a study like this is the potential to compare fine- and coarse scale temporal and spatial temperature variability in predicting bleaching. While the study does a nice job of isolating the Daily Temperature Range (DTR) as a major predictor at the reef scale (<1km, daily), I think there is a missed opportunity here to compare in situ temperature metrics with satellite 4km biweekly SST data (which is globally available) for bleaching predictions which would be more useful from a management standpoint. It is not novel to show that more thermally variable sites have more thermally tolerant corals, but this analysis could be easily extended using the same model framework to provide a valuable comparison across scales. This is my major recommendation.

Line 79-81, Lines 84-85, Lines 90-92. It would be a great contribution if the authors could include the MMM climatology (and perhaps MMMmax; Donner et al. 2011) calculated based on 4km satellite SSTs in their model to compare how remotely sensed data compares with the in situ data predictors. You do not necessarily need the NOAA CRW virtual stations for this; you can

obtain the data from CoRTAD (<https://data.noaa.gov/dataset/the-coral-reef-temperature-anomaly-database-cortad-version-5-global-4-km-sea-surface-temperature07731>). This could be especially useful for locations that do not have *in situ* data available. This comment also applies to the categories of “Background Conditions”, “Cumulative Thermal Stress” and “Rate of Heating”. The kind of analysis performed here could also be very useful for improving bleaching predictions at coarser spatial scales especially because among the highest-ranking logit models, cumulative (and acute) thermal stress largely contributed to bleaching (Lines 188-190).

We are thankful to Reviewer #3 for directing our attention to the CoRTAD dataset, so that we could examine how the calculated values of our thermal metrics differed when they were computed using weekly 4 km SST as opposed to *in situ* temperature data. We obtained CoRTAD temperature data from the pixels closest to our *in situ* locations, and proceeded to calculate 4 km versions of MMM, MMM_{Max}, Acute1, Acute2, DHW, ROTC_{SS}, and ROTC₉₀ (in this passage we will refer to the *in situ* or 4 km versions of these variables by adding the appropriate subscript, i.e. MMM_{*in situ*} or MMM_{4 km}). Note that we did not compute MMM_{Max-*in situ*} because many of our time series were just over 1 year in duration, and therefore MMM_{Max-*in situ*} would be interchangeable with MMM_{*in situ*}. Furthermore, we did not compute DTR_{30-4 km} because of the temporal limitations of the CoRTAD sampling.

Below we compare the *in situ* and remote sensed computations of various thermal metrics. Here we see that MMM and MMM_{Max} yield relatively strong correlations when comparing their *in situ* and 4 km versions ($r = 0.92$ and 0.90 , respectively; $p \ll 0.05$ for both). The binary-valued Acute1_{*in situ*} agrees with Acute1_{4 km} 66.7% of the time within our dataset, while Acute2_{*in situ*} agrees with Acute2_{4 km} 76.5% of the time, although many of these observations are equal to 0 (‘No Stress’), which may trivialize the agreement. The 4 km DHW product shows a significant correlation with its *in situ* counterpart, though this correlation is relatively weak ($r = 0.67$; $p \ll 0.05$), which may be explained through their different temporal resolutions (the *in situ* data has higher temporal resolution). Finally, ROTC_{SS-*in situ*} and ROTC_{90-*in situ*}, which both have significant positive correlations with ROTC_{SS-4 km} and ROTC_{90-4 km}, respectively, still have weak r values (0.67 and 0.23 respectively). This necessitates an investigation of how these 4 km thermal metrics perform within our model framework compared to the *in situ* versions reported in the Main Text.

After computing CoRTAD-derived covariates and fitting OLR models, we found that the inclusion of 4 km versions of MMM, MMM_{Max}, DHW, and ROTC, when coupled with Depth and DTR (only available from *in situ* data), significantly improved model performance.

- The following passage has been added to the **Results** section of the Main Text (Lines 212-233): “To broaden the applicability of our conclusions, we repeated the OLR analysis, with the addition of remotely sensed SST-derived covariates, to determine how the results would differ from the *in situ* driven models. We obtained weekly 4 km resolution CoRTAD SST data⁵³, using the data pixels closest to the coordinates of our *in situ* loggers, and used this SST dataset to compute covariates within the Background Conditions, Acute and Cumulative Thermal Stress, and Heating Rate categories (Table 1). This resulted in an improved highest ranked model (*Methods*), that included six covariates, three of which (MMM, DHW, Rate of Temperature Change) were computed using the SST, as opposed to *in situ*, data (Figure 5a). However, similar to the highest-ranked model fit to exclusively *in situ* data, covariates representing High-Frequency Temperature Variability, specifically DTR₃₀, and depth were again the dominant drivers of bleaching, and served as mitigating factors (Figure 5a,b). Similarly, amongst covariates that exacerbated bleaching, Acute and Cumulative Thermal Stress provided the strongest influence (Figure 5a,b), while Background Conditions (MMM_{4 km}) represented a mild exacerbating effect. A notable difference occurring in these new models is the opposite effect Heating Rate has from before; whereas in the exclusively *in situ* models, Heating Rate exacerbated bleaching, these SST-based models imply stronger heating rates serve to mitigate bleaching. Ultimately, these SST-based OLR models indicate that upon consideration of the consistent importance of daily temperature range to bleaching, a globally available remotely sensed metric for diurnal temperature variability would be valuable for improved bleaching predictions”

- The following passage has been added to the **Ordinal Logistic Regression** section of the *Methods* section of the Main Text and highlighted: “The SST-based OLR analysis summarized in Figure 5 was performed using the CoRTAD 4 km weekly SST data⁵³ pixels that were closest to our *in situ* bleaching observations. The quantities MMM, MMM_{Max}³⁰, Acute1, Acute2, DHW, ROTC_{SS}, and ROTC₉₀ (Table 1) were computed from the CoRTAD temperature data. This resulted in a total of 27 covariates (20 as described above, and 7 new ones computed from the CoRTAD data). We then proceeded to fit OLR models using all permutations of covariates, with the constraint that within each model, we only include ≤ 1 covariate from each category listed in Table 1. This produced 60,479 models, from which we performed model comparison using AIC_C, resulting in 12 dominant models being identified as statistically indistinguishable. Common covariates in these models were MMM, MMM_{Max}, MMM_{4 km}, MMM_{Max-4 km}, DHW₃₀, DHW_{4 km}, Acute1, Acute2_{4 km}, and ROTC_{90-4 km}, and each of these 12 highest-ranked models included at least 1 SST-based covariate. The AIC_C values of these models ranged from 129.19 to 131.12, with McFadden’s pseudo-R² values ranging from 0.35 to 0.38, which presents a considerable improvement to the highest-ranked model that was fit to only *in situ* values, which had an AIC_C equal to 143.75 and a McFadden’s pseudo-R² of 0.30. To highlight the importance of High-Frequency Temperature Variability in model improvement, we also fit these models to the data and excluded the Daily Temperature Range covariate, which was either DTR₃₀ or DTR₉₀. This significantly decreased model performance, with AIC_C values ranging from 153.46 to 169.57, and McFadden’s pseudo-R² values ranging from 0.12 to 0.23, which supports the idea that DTR₃₀ is the dominant driving variable for the bleaching response.”

Figure 5 | Remotely Sensed SST OLR Results. a) Parameter estimates for standardized model coefficients of the covariates used in the highest-ranked OLR model when weekly 4 km CoRTAD SST-based variables are added to the pool of possible covariates; standard errorbars were computed from delete-1 jackknife resampling. b) The summation of the standardized covariate coefficients grouped by category from the highest-ranked models when including CoRTAD SST-based covariates; the standard errorbars shown are obtained by summing in quadrature the individual standard errors computed after delete-1 jackknife resampling. The SST-based covariate contribution to each Cumulative Effect is colored grey.

- The 4 km variables have also been added to Table 1.
- The previous figure labeled Figure 5 (Case Studies) has now been relabeled to Figure 6.

Lines 43-62. It might be worth noting that other modelling studies at the global scale have also shown that including coarser scale temperature variability in the calculation of the SST climatology improves bleaching predictions:

Boylan, P., & Kleypas, J. (2008). New insights into the exposure and sensitivity of coral reefs to ocean warming. In Proc 11th Int Coral Reef Symp (Vol. 2, pp. 854-858).

Donner, S. D. (2011). An evaluation of the effect of recent temperature variability on the prediction of coral bleaching events. *Ecological Applications*, 21(5), 1718-1730.

Logan, C. A., Dunne, J. P., Eakin, C. M., & Donner, S. D. (2012, July). A framework for comparing coral bleaching thresholds. In Proceedings of the 12th International Coral Reef Symposium (ed Yellowlees D, Hughes TP), pp 10A3 Townsville [Internet].

Thank you for introducing us to these three references. We are already citing Donner (2011) in the Main Text where we speak of using MMM_{Max} in our CoRTAD 4 km SST OLR models. As for lines 43-62, we will cite Donner (2011), Logan et al (2012), and Boylan & Kleypas (2008), and have changed the first lines of that passage to:

“Bleaching predictions from remotely sensed temperatures can be improved even through the calculation of temporally coarse interannual temperature variability^{30,31} and coral sensitivity to thermal stress exposure³². However, site-specific studies suggest that historical temperature variability within diurnal time scales affects corals’ physiological tolerance^{16,23,33,34} and performance³⁵ under thermal stress”.

Lines 74-105. I suggest breaking this paragraph up to make it more read-able. Perhaps you could discuss the categories that include a measure of high frequency variability in a 2nd paragraph, and the PCA analysis in a 3rd. Also I am only seeing 7/8 categories listed in this paragraph. I think you’ve accidentally omitted ‘Acute thermal stress’.

The passage was split into 2 paragraphs, where the second paragraph begins the discussion on the PCA.

At the time, we did not feel it was necessary to emphasize the Acute Thermal Stress category, but for the sake of harmony in that paragraph, the line “The ‘Acute Thermal Stress’ category was included as a safeguard to differentiate sites with temperatures that may not have exceeded $MMM + 1\text{ }^{\circ}\text{C}$ (i.e. no thermal stress) yet still experienced bleaching,” was added.

Line 76-77. Is reef type included as a variable (e.g. fore vs back reef)? If not, could you comment on this?

We initially tried making a quantitative ordinal variable for reef type based on distance from shore, however in some instances, there was too much subjectivity in differentiating habitat terminology (i.e. a back reef vs a lagoon vs a back reef lagoon) within our data set. We

nonetheless included a trial version of this variable in our OLR models, using the values 1-3 to represent back reefs, reef flats, and reef slopes/crests, respectively. This trial covariate did not appear in any of the highest ranked models, and we chose to formally preclude it from the analysis. Furthermore, we kindly refer Reviewer#3 to Supplementary Figure 2, which implies that other thermal metrics already capture the effects of habitat. This in turn implies that if we did include habitat as a variable, it may result in another nonlinear interaction which we would have to account for.

- To the *Methods* section of the Main Text, we added the lines “We could not concisely define habitat (i.e. strictly reef flat, reef crest, or reef slope, etc) information at which many loggers were placed, and therefore did not include habitat as an explanatory variable.” and highlighted these changes.

Lines 221. Omit “within coral genes” as it is not necessary.
This change was made.

Line 227. Define what is meant by “encourage”. Can you comment on what specific mechanism(s) you are referring to (acclimatization, community shifts, adaptation)?
Thank you for helping us clarify our language. In this specific instance, based on the references cited, we were referring to the mechanisms of adaptation or acclimation and acclimatization, and using the phrase “encourage greater tolerance to thermal stress” helped explain the symptom we were trying to describe. We added the words “(via acclimation or adaptation)” to that sentence.

Lines 228-230. Globally, it is hard to disentangle if similar bleaching patterns across regions are a result of community shifts towards more tolerant species over time vs acclimatization or natural selection. I think the discussion warrants further discussion of how which mechanisms could lead the observed findings across regions.

Reviewer #3 raises an important point, and we completely agree that there is a species-dependence to observed bleaching responses, both within and across regions. However, here we are at an impasse due to a lack of taxon-specific observations. Reviewer #2 also noted this issue, and we inserted a few lines to the **Discussion** in an attempt to address that. We return to Swain et al (2016), which provides a relevant understanding that differential bleaching responses may be a result of intrinsic factors (holobiont biology), extrinsic factors (environmental conditions), or measurement uncertainty. That study concludes that while bleaching heterogeneity may be explained both by intrinsic and extrinsic factors within a reef site, coral intrinsic factors dominate the bleaching response across regions. Within our Main Text, regarding the sentence in question (“Further, our results establish that the resistance of corals located in areas of high-frequency temperature variability to bleaching occurs in reef regions throughout the world...”) however, we are simply stating that diurnal temperature variability corresponds with a lower incidence of bleaching, without making any assumptions of taxon-specific interaction with diurnal temperature variability. A more thorough discussion of which mechanisms (depth, background conditions, acute/cumulative thermal stress, high-frequency temperature variability, etc), would require more site-specific analysis coupled with species-level bleaching observations both within and across reefs, which we currently cannot accommodate for.

- The following lines were inserted into the **Discussion** of the Main Text (Lines 263-267):
“While we lack sufficient species-level data, we fully acknowledge that intrinsic coral properties³⁹, differences in reef-scale community compositions⁵³, and taxonomic susceptibility²³ are likely to influence heterogeneous reef-scale bleaching responses, and an improvement to our model framework would include species-level covariates.”

Table 1. Under the Category 5 row, ‘stress’ is mis-spelled.
The error was corrected.

Table 1. Perhaps re-order the categories into those that include a measure of “high frequency variability” (e.g., 4, 7, 8) vs those that do not (1,2, 3?, 5). I had a hard time teasing apart which metrics required daily temperatures to calculate (vs those that could potentially be calculated using satellite SSTs).

The choice of ordering these categories was slightly arbitrary; it seems initially we tried to introduce the most ubiquitous, accessible, and well known variables first, and then proceeded to describe the more computationally involved, recently introduced, or obscure variables. Unless Reviewer #3 feels very strongly about this, we would like to keep the ordering of these the same, as behind the scenes, many of our analyses relies on this ordering of categories.

Reviewers' comments:

Reviewer #1 (Remarks to the Author):

The authors have done a very good job taking all reviewers' comments on board and revising their manuscript accordingly.

I am satisfied with the revised version and now happy to recommend it for publication.

Reviewer #2 (Remarks to the Author):

For the most part, the authors have addressed the issues that concerned me, and the additional analyses suggested by others have measurably improved the paper. My one remaining concern centers on the spectral analyses – I believe they should find a way to present analyses that do not have false spectral peaks, rather than just stating that some peaks are likely false. The conclusions regarding subdaily variability (a critical part of their analysis) will be much strengthened (see comments below on lines 134-143).

Line by line edits are listed below:

Lines 66-68 and 107-109 feel redundant.

Line 134-143: It appears that the authors are using the spectral band that includes methodological artifacts as part of a metric to define high frequency variability. They certainly need to revise that approach to be sure they aren't including these false peaks. There are many types of spectral methods, and it would be much cleaner if the authors could find a method that avoids false peaks. Regarding attribution of the higher (sub-diurnal) variability, I see the logic of including tides but not wind and waves, unless there is a specific mechanism for these to create periodic variance in SST on the appropriate scale (ca. 1-12 hours).

Lines 230-249 (new paragraph) seems to have capitalization issues – I'll leave that to the editor.

Lines 257-258 This sentence is not very satisfying – presumably all the reefs have environmental data available or they would not be in this study (?).

Line 294: "that MAY arise"- ?? How about "that is NOW arising."

Line 299: "events" implies a specific event, but I think the authors are referring instead to a "regime" of more frequent/intense bleaching.

Line 311 publicly available. Source URLs should be given.

Line 327 extremes.

Reviewer #3 (Remarks to the Author):

The authors have done an excellent job of extensively responding to my own (and other reviewer's) comments. Inclusion of remotely sensed SST data into the model significantly increases the impact of this study and I appreciate the author's including the additional analysis. I have no other comments, and recommend the manuscript for publication.

Point-by-point response to comments on “High-frequency temperature variability reduces the risk of coral bleaching.”

September 2017

Reviewers' comments in black; Author responses in blue text.

Reviewers' comments:

Reviewer #1 (Remarks to the Author):

The authors have done a very good job taking all reviewers' comments on board and revising their manuscript accordingly. I am satisfied with the revised version and now happy to recommend it for publication.

Reviewer #2 (Remarks to the Author):

For the most part, the authors have addressed the issues that concerned me, and the additional analyses suggested by others have measurably improved the paper. My one remaining concern centers on the spectral analyses – I believe they should find a way to present analyses that do not have false spectral peaks, rather than just stating that some peaks are likely false. The conclusions regarding subdaily variability (a critical part of their analysis) will be much strengthened (see comments below on lines 134-143).

We thank Reviewer #2 for compelling us to provide a more thorough accounting of the spectral peaks seen in Figure 2a and Supplementary Figure 1. After revisiting the methods we've used to produce these plots of power spectral density (PSD) and identify their statistically significant spectral peaks, we are confident that the subdiurnal (corresponding to periods less than a day) spectral peaks we have indicated in the figures in question are not artifacts of the spectral method. Focusing on the time series shown in Figure 2a, we used various subsampling, interpolating, and ensemble averaging methods, as well as different window functions and sizes, to compute spectral density estimates for the time series, and yet the peaks in question were still present.

Therefore, we are left with three likely explanations for these peaks. The first is that these subdiurnal peaks are indeed physically present at the sites in question and are overtides caused by the nonlinear interaction of lower frequency tidal constituents with the shallow bathymetry. Another plausible explanation is that aliasing due to undersampling can fold spectral energy from higher frequencies into lower ones. In order to resolve peaks at 3, 4, 5, 6, and 7 cycles per day in the frequency domain, the corresponding required sampling intervals are 4, 3, 2.4, 2, and 1.7 hours, respectively. For the time series presented in Figure 2, and frankly, for the overwhelming majority of time series in this study, sampling intervals were less than or equal to 1 hour, and therefore, these frequencies corresponding to subdiurnal periods are theoretically resolvable. Therefore, unless there are considerably strong periodic high frequency components inherent in these time series at regularly spaced frequencies greater than ~ 12 cpd (or one-half

times our typical sampling frequency), it is doubtful that the subdiurnal peaks exhibited in our study are due solely to insufficient sampling and hence aliasing.

A third possible explanation is that these subdiurnal peaks are present because tidal asymmetry, or the unequal durations of rising and falling water levels or unequal peak ebb and flood currents, leads to asymmetry in heating and cooling in the temperature time series (Speer and Aubrey 1985; Friedrichs and Aubrey 1988). This asymmetry in the ebb and flood current magnitudes may arise from density-driven circulation, topography and bathymetry effects, or nonlinear interactions amongst tidal constituents (Godin 1983), such that various astronomical constituents interact to produce compound overtides (Parker 1991, Wang et al 1999, Song et al 2010, Guo et al 2015). A wealth of studies (i.e. Godin 1983, Aubrey and Speer 1985, Speer and Aubrey 1985, Friedrichs and Aubrey 1988, Speer et al 1991, Song et al 2010, Guo et al 2015) describes how asymmetries between the ebb and flood contributions to harmonic components within a signal are responsible for transferring spectral energy from low frequencies to higher ones. Tidal asymmetry, therefore, can alias spectral energy from the primary tidal constituents into higher frequency harmonics, for example, such that M2 energy can be aliased to the M4, M6, and M8 shallow water overtides (Pugh, 1987; Parker, 1991; Le Provost, 1991; Walters and Werner, 1991; Wang et al., 1999). Furthermore, this aliasing effect can produce significant spectral energy at frequencies that are not present in the original oceanographic time series, even at nearly 5 and 7 cpd (Polagye et al 2010). As we know that many of the temperature time series used in this study come from shallow coastal sites with highly nonlinear interactions between bottom friction, turbulence, and tidal constituents, it is likely that perhaps some of the subdiurnal peaks seen in our power spectral density estimates are due to tidal asymmetry.

While we have provided our three best hypotheses as to why these subdiurnal peaks appear in some of our power spectral density estimates, we still cannot specify whether these peaks are physically present in the underlying data, or whether they are due to the harmonic transfer of spectral energy. A rigorous accounting of these higher frequency peaks is perhaps beyond the scope of this paper, and ultimately, the ordinal logistic regression results are not dependent on the spectral analysis, as the calculation of daily temperature ranges within a temperature time series does not depend on these subdiurnal peaks. Therefore, to address the concerns of Reviewer #2 and not present peaks which we find suspicious (i.e. perhaps not representative of an underlying physical process in the data, or due to tidal asymmetry), we have modified our computation of the PSD estimates in our study so as to present spectral density estimates for frequencies only up to 4 cpd. To do this, we subsampled all temperature time series to a 3 hour sampling interval, which is consistent with our quality control criteria, so as to only resolve frequencies of up to 4 cpd. Note however, that in doing so, we have possibly shifted spectral energy from higher frequencies into frequencies ≤ 4 cpd, although, as we will see, this did not alter the results in any operable manner. This resulted in the following changes to the Main Text and Supplementary Information:

The following words were removed from Line 128 of the Main Text:

- , though there may be harmonic contributions to the spectral peaks occurring on 3–4 cpd frequencies, an artifact of the spectral technique.

Lines 323-325 of the **Spectral Analysis** paragraph of the **Methods** section in the Main Text were changed to:

- Power spectral density (PSD) estimates were computed for each temperature time series. First, if necessary, temperature time series were resampled or linearly interpolated to maintain a constant sampling interval, chosen to be 3 hours so as to resolve spectral frequencies of up to 4 cpd.

Figure 2 is now:

Figure 2 | Temperature variability of six reef records. (a) Power spectra of temperature with asterisks marking significant peaks, (b) yearly composites of mean daily temperatures and temperature ranges (red and pink shading, respectively), (c) 7-day trends in temperatures at two different habitats on the reef, and (d) histograms of daily temperature range at the same two habitats on each reef. In each case, reef locations are shown in maps at the left (for site information see Supplementary Table 1), the full duration of temperature records are indicated in (a), and the great-circle distances between same-reef sites are indicated in (d).

Supplementary Figure 1 is now:

Supplementary Figure 1| Power spectra at high-frequencies. Power Spectral Density (PSD) estimates of water temperature from every time series, focused on the high frequency (0.727 to 4 cpd) part of the spectrum. PSD estimates are sequentially offset by $10^{-0.2} \text{ }^{\circ}\text{C}^2/\text{cpd}$, and significant peaks are marked with an asterisk. PSD estimates are identified by their 3-letter abbreviations (Supplementary Table 1) and color-grouped according to oceanographic region. Frequencies, in cycles per day (cpd) of important astronomical tidal constituents are indicated at the top.

And the 1st paragraph of the Supplementary Information was changed to:

- Power Spectral Density (PSD) estimates of the 118 temperature time series reveal that all had significant diurnal and semidiurnal spectral peaks, emphasizing the ubiquity of significant high-frequency variance throughout our data set (Supplementary Figure 1). The majority of time series also showed significant peaks corresponding to the shallow water terdiurnal and quarterdiurnal frequencies (99.2% and 92.4% of time series, respectively). Finally, 43 (30.5%) of the time series showed significant spectral peaks corresponding to long term, annual frequencies (1/90 – 1/540 cpd). Statistically significant spectral peaks are indicated in Supplementary Figure 1 to illustrate the persistence of high frequency water temperature variability among many different types of coral reef environments. Regardless of oceanographic region or latitudinal gradient, there is a considerable amount of temperature variability within diurnal and shorter periods that corals are exposed to.

Supplementary Figure 2 is now:

Supplementary Figure 2| Ratios of high-frequency to seasonal variance by habitat. High-frequency (1/33 to 1/6 h) and seasonal (1/7 to 1/84 d) variance were computed via integration of the power spectral densities within these respective frequencies. Histograms of the ratio of these two variances are shown grouped by reef habitat, and the results of a Kruskal-Wallis test are shown in the inset text.

Lines 132-135 of the Main Text are now:

- At back reef, reef flat, and reef slope habitats, these ratios were on average 1.83, 0.68, and 0.44, respectively, while across all locations, this ratio was 1.02 (Supplementary Fig. 2a). Furthermore, these ratios differed significantly among the three habitats ($\chi^2 = 24.66$, df = 117, $p \ll 0.05$; Supplementary Fig. 2b).

References:

- Friedrichs, Carl T., and David G. Aubrey. "Non-linear tidal distortion in shallow well-mixed estuaries: a synthesis." *Estuarine, Coastal and Shelf Science* 27.5 (1988): 521-545.
- Godin, Gabriel. "The spectra of point measurements of currents: their features and their interpretation." *Atmosphere-Ocean* 21.3 (1983): 263-284.
- Guo, Leicheng, et al. "River - tide dynamics: Exploration of nonstationary and nonlinear tidal behavior in the Yangtze River estuary." *Journal of Geophysical Research: Oceans* 120.5 (2015): 3499-3521.
- Le Provost, Christian. "Generation of overtides and compound tides." *Tidal hydrodynamics* (1991): 269-296.
- Parker, Bruce B. *Tidal hydrodynamics*. John Wiley & Sons, 1991.
- Polagye, B. L., Jeff Epler, and Jim Thomson. "Limits to the predictability of tidal current energy." *OCEANS 2010*. IEEE, 2010.
- Pugh, D. T. "Tides, surges and mean sea-level: a handbook for engineers and scientists, 472 pp." (1987).
- Song, Dehai, et al. "The contribution to tidal asymmetry by different combinations of tidal constituents." *Journal of Geophysical Research: Oceans* 116.C12 (2011).

Speer, P. E., and D. G. Aubrey. "A study of non-linear tidal propagation in shallow inlet/estuarine systems Part II: Theory." *Estuarine, Coastal and Shelf Science* 21.2 (1985): 207-224.

Walters, Roy A., and Francisco E. Werner. "Nonlinear generation of overtides, compound tides, and residuals." *Tidal hydrodynamics* (1991): 297-320.

Wang, Z. B., C. Jeuken, and H. J. De Vriend. *Tidal asymmetry and residual sediment transport in estuaries*. Deltares (WL), 1999.

Line by line edits are listed below:

Lines 66-68 and 107-109 feel redundant.

Lines 106-109 were changed to:

In addition to the *in situ* variables, we include 7 analogous and conventional remotely sensed SST metrics (Table 1). After standardizing all covariates and fitting them to ordinal-valued bleaching prevalence scores using ordinal logistic regression (OLR) models (*Methods*), we conclude that

Line 134-143: It appears that the authors are using the spectral band that includes methodological artifacts as part of a metric to define high frequency variability. They certainly need to revise that approach to be sure they aren't including these false peaks. There are many types of spectral methods, and it would be much cleaner if the authors could find a method that avoids false peaks. Regarding attribution of the higher (sub-diurnal) variability, I see the logic of including tides but not wind and waves, unless there is a specific mechanism for these to create periodic variance in SST on the appropriate scale (ca. 1-12 hours).

The text and methods were changed to avoid showing spectral artifacts (see above). The words 'and higher-frequency' were removed, to focus the passage solely on diurnal mechanisms.

Lines 230-249 (new paragraph) seems to have capitalization issues – I'll leave that to the editor. Thank you for pointing this out. This indeed is perhaps an issue we must follow up on with the editor.

Lines 257-258 This sentence is not very satisfying – presumably all the reefs have environmental data available or they would not be in this study (?).

We are not sure which parts of Lines 257-258 Reviewer #2 is referring to; perhaps instead the Reviewer is referring to Line 249? In that case, we have changed the sentence to read "...due to the availability of additional meteorological data..."

Line 294: "that MAY arise"- ?? How about "that is NOW arising."

The change was made.

Line 299: "events" implies a specific event, but I think the authors are referring instead to a "regime" of more frequent/intense bleaching.

The change was made.

Line 311 publicly available. Source URLs should be given.

The hyphen was removed. Source URLs and references are given in the Supplementary Information and Supplementary Table 1.

Line 327 extremes.

The change was made.

Reviewer #3 (Remarks to the Author):

The authors have done an excellent job of extensively responding to my own (and other reviewer's) comments. Inclusion of remotely sensed SST data into the model significantly increases the impact of this study and I appreciate the author's including the additional analysis. I have no other comments, and recommend the manuscript for publication.

REVIEWERS' COMMENTS:

Reviewer #2 (Remarks to the Author):

I appreciate the authors' thorough treatment of the high frequency issues I raised, and I am satisfied with the resulting changes in the manuscript.

Just one comment - I notice in the revised figure 2, lowest row, that despite strong subdaily spectral peaks, the SST time series show virtually no visually apparent variability at the subdaily scale, suggesting its not "real" (or not really substantial). If further exploration of this issue were desirable, they might consider filtering the data at subdaily periods to see whether the scale of variance exceeded the measurement uncertainty of the thermometers, or whether it is strongest at certain tidal phases. But for this paper, that is a distraction - the focus on the daily cycle is the key point and I am satisfied with the way the authors have modified the manuscript for publication.

Point-by-point responses to comments on “High-frequency temperature variability reduces the risk of coral bleaching.”

March 2018

Reviewers' comments are in black text; Author responses are in blue text.

Reviewers' comments:

Reviewer #2 (Remarks to the Author):

I appreciate the authors' thorough treatment of the high frequency issues I raised, and I am satisfied with the resulting changes in the manuscript.

Just one comment - I notice in the revised figure 2, lowest row, that despite strong subdaily spectral peaks, the SST time series show virtually no visually apparent variability at the subdaily scale, suggesting its not "real" (or not really substantial). If further exploration of this issue were desirable, they might consider filtering the data at subdaily periods to see whether the scale of variance exceeded the measurement uncertainty of the thermometers, or whether it is strongest at certain tidal phases. But for this paper, that is a distraction - the focus on the daily cycle is the key point and I am satisfied with the way the authors have modified the manuscript for publication.

We are pleased to hear that Reviewer #2 is satisfied with the current version of the manuscript, and we again thank Reviewer #2 (as well as Reviewers #1 and #3) for helping improve the quality of this manuscript. Their comments have certainly focused our attention to crucial items that we overlooked during the drafting of this manuscript, and we sincerely appreciate the help of all Reviewers.

Regarding the temperature time series presented in Figure 2, the PSD estimates shown in column **a** are for the time series from the more shallower or shoreward sites of each pair; these are indicated by red letters in the map, and red plotting lines in columns **b-d**. Therefore, the PSD estimate seen in the last row of Fig. 2a is for HR1, which does indeed appear to show considerable diurnal temperature variability in Fig. 2c (red line). In order to reflect this more appropriately, we have changed the Figure legend to read:

Figure 2 | Temperature variability of six reef records. (a) Power spectra of temperature for TA3, P21, OF3, VT1, HW1, and HR1, with asterisks marking significant peaks, (b) yearly composites of mean daily temperatures and temperature ranges (red and pink shading respectively) for the same 6 time series in **a**, (c) 7-day trends in temperatures at two different habitats on the reef, and (d) histograms of daily temperature range at the same two habitats on each reef. In each case, reef locations are shown in maps on the left (for site information see Supplementary Table 1), the full duration of temperature records are indicated in (a), and the great-circle distances between same-reef sites are indicated in (d).

We hope this addresses the issue that Reviewer #2 raises, and we also hope we are not providing an overly simple explanation to the issue. We thank Reviewer #2 for outlining a more rigorous analysis to assess the subdiurnal variability for the time series in question, and we may decide to try this analysis at a future time.